# Robust Amortized Bayesian Inference with Self-Consistency Losses on Unlabeled Data

**Aayush Mishra**[*]
Department of Statistics
TU Dortmund University, Germany
aayush.mishra@tu-dortmund.de

**Daniel Habermann**[*]
Department of Statistics
TU Dortmund University, Germany
daniel.habermann@tu-dortmund.de

**Marvin Schmitt**
Independent Scientist
marvinschmitt@gmail.com

**Stefan T. Radev**
Department of Cognitive Science
Rensselaer Polytechnic Institute, USA
radevs@rpi.edu

**Paul-Christian Bürkner**
Department of Statistics
TU Dortmund University, Germany
paul.buerkner@gmail.com

## Abstract

Amortized Bayesian inference (ABI) with neural networks can solve probabilistic inverse problems orders of magnitude faster than classical methods. However, ABI is not yet sufficiently robust for widespread and safe application. When performing inference on observations outside the scope of the simulated training data, posterior approximations are likely to become highly biased, which cannot be corrected by additional simulations due to the bad pre-asymptotic behavior of current neural posterior estimators. In this paper, we propose a semi-supervised approach that enables training not only on labeled simulated data generated from the model, but also on *unlabeled* data originating from any source, including real data. To achieve this, we leverage Bayesian self-consistency properties that can be transformed into strictly proper losses that do not require knowledge of ground-truth parameters. We test our approach on several real-world case studies, including applications to high-dimensional time-series and image data. Our results show that semi-supervised learning with unlabeled data drastically improves the robustness of ABI in the out-of-simulation regime. Notably, inference remains accurate even when evaluated on observations far away from the labeled and unlabeled data seen during training. The code is available at https://github.com/bayesflow-org/self-consistency-real.

## 1 Introduction

Theory-driven computational models (mechanistic models) are highly influential across numerous branches of science (Lavin et al., 2021). The utility of computational models largely stems from their ability to fit real data $x$ and extract information about hidden parameters $\theta$. Bayesian methods have been instrumental for this task, providing a principled framework for uncertainty quantification and inference (Gelman et al., 2013). However, gold-standard Bayesian methods, such as Gibbs or Hamiltonian Monte Carlo samplers (Brooks et al., 2011), remain notoriously slow. Moreover, these methods are rarely feasible for fitting complex models (Dax et al., 2021) or even simpler models in big data settings with many thousands of data points in a single dataset (Blei et al., 2017), or when thousands of independent datasets require repeated model re-fits (von Krause et al., 2022).

---

[*]Equal contribution

In recent years, deep learning methods have helped address some of these efficiency challenges (Cranmer et al., 2020). In particular, *amortized Bayesian inference* (ABI; Gershman & Goodman, 2014; Le et al., 2017; Gonçalves et al., 2020; Radev et al., 2023; Gloeckler et al., 2023; Elsemüller et al., 2024; Zammit-Mangion et al., 2024) has received considerable attention for its potential to automate Bayesian workflows by replacing per-dataset inference with a learned inference model. By training generative neural networks on simulations generated from a probabilistic model, ABI learns a mapping from observations to posterior distributions, allowing subsequent inference on real data to be performed in near-instant time. This amortization trades an upfront computational cost during training for substantial gains in scalability and reusability, making Bayesian inference feasible in settings involving large datasets, intractable likelihood, or real-time constraints.

However, due to the reliance on pre-trained neural networks, ABI methods can become unreliable when applied to data that is unseen or sparsely encountered during training. In particular, posterior samples from amortized methods may deviate significantly from samples obtained with gold-standard MCMC samplers when there is a mismatch between the simulated training data and the real data (Ward et al., 2022; Schmitt et al., 2023; Siahkoohi et al., 2023; Gloeckler et al., 2023; Frazier et al., 2024). This lack of robustness limits the widespread and safe applicability of ABI methods.

In this work, we propose a new *robust semi-supervised approach to ABI*. The supervised part learns from a "labeled" set of parameters and corresponding synthetic (simulated) observations, $\{\theta, x\}$, while the unsupervised part leverages an "unlabeled" data set of real observations $\{x^*\}$ without parameters. In contrast to other methods aiming to enhance the robustness of ABI, our approach does not require ground-truth parameters $\theta^*$ (Wehenkel et al., 2024), post hoc corrections (Ward et al., 2022; Siahkoohi et al., 2023), or specific adversarial defenses (Gloeckler et al., 2023), nor does it entail a loss of amortization (Ward et al., 2022; Huang et al., 2023) or generalized Bayesian inference (Gao et al., 2023; Pacchiardi et al., 2024).

While Schmitt et al. (2024) introduced self-consistency (SC) losses to improve simulation efficiency in a fully supervised, simulation-only setting, we extend self-consistency losses to unsupervised training on real observations, enabling robust inference without access to ground-truth parameters. We prove that self-consistency losses are strictly proper, target the true analytic posterior, and are agnostic to the data distribution. These properties allow them to be seamlessly integrated with any simulation-based loss for semi-supervised training without introducing a trade-off in the training objective. Empirical results on diverse tasks, including high-dimensional time series and images, show that our method preserves ABI's characteristic speed while achieving strong robustness, even with as few as four unlabeled real-world observations. Posterior estimates remain accurate and well-calibrated, generalizing beyond both labeled and unlabeled training data.

## 2 METHODS

### 2.1 BAYESIAN SELF-CONSISTENCY

Self-consistency leverages a simple symmetry in Bayes' rule to enforce more accurate posterior estimation even in regions with sparse data (Schmitt et al., 2024; Ivanova et al., 2024). Crucially, it incorporates the likelihood (when available) or a surrogate likelihood during training, thereby providing the networks with *additional information* beyond the standard simulation-based loss typically employed in ABI (see below).

Under exact inference, the marginal likelihood is independent of the parameters $\theta$. That is, the Bayesian self-consistency ratio of likelihood-prior product and posterior is constant across any set of parameter values $\theta^{(1)}, \ldots, \theta^{(L)}$,

$$p(x) = \frac{p(x \mid \theta^{(1)}) \, p(\theta^{(1)})}{p(\theta^{(1)} \mid x)} = \cdots = \frac{p(x \mid \theta^{(L)}) \, p(\theta^{(L)})}{p(\theta^{(L)} \mid x)}. \tag{1}$$

However, replacing $p(\theta \mid x)$ with a neural estimator $q(\theta \mid x)$ (and analogously for the likelihood) leads to undesired variance in the marginal likelihood estimates across different parameter values on the right-hand side. Since this variance is a proxy for *approximation error*, we can directly minimize it via backpropagation along with any other ABI loss to provide further training signal and reduce errors guided by density information. Our proposed semi-supervised approach builds on these advantageous properties.

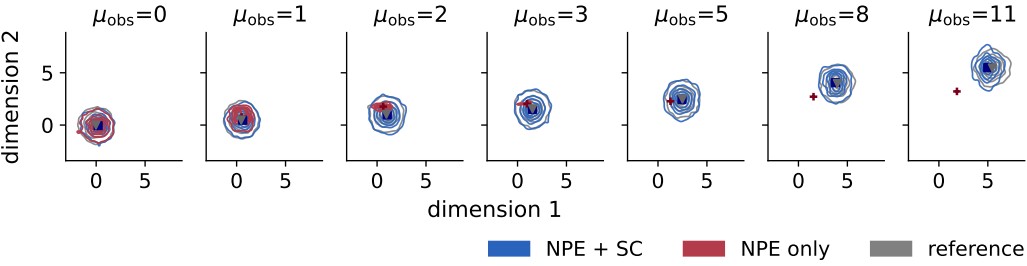

Figure 1: Contour plot of the normal means problem using standard NPE (red) or our semi-supervised approach (NPE + SC, blue), with the analytic posterior in gray. Symbols indicate posterior mean estimates (red cross: NPE only; blue square: NPE + SC; gray triangle: reference). Each subplot shows posterior inference on observed data that are increasingly distant from the labeled training data ($\mu_{\mathrm{prior}} = 0$). Only the first two dimensions of the 10-dimensional posterior are shown. While standard NPE collapses to zero variance for $\mu_{\mathrm{obs}} \geq 2$, adding the self-consistency loss preserves accurate posterior estimates even far beyond both training spaces ($\mu_{\mathrm{obs}} > 3$). Training was performed using the default configuration (see Section 4.1).

## 2.2 SEMI-SUPERVISED AMORTIZED BAYESIAN INFERENCE

The marginal likelihood identity in Eq. (1) is straightforward, but practically never used in traditional sampling-based methods (e.g., MCMC) because they do not provide a closed-form for the approximate posterior density $q(\theta \mid x)$. In contrast, we can readily evaluate $q(\theta \mid x)$ in ABI when using a neural density estimator that allows efficient density computation (e.g., normalizing flows, Kobyzev et al. (2020)). Thus, we can formulate a family of *semi-supervised losses* of the form:

$$(q^*, h^*) = \underset{q,h}{\mathrm{argmin}}\, \mathbb{E}_{(\theta,x)\sim p(\theta,x)} \left[ S(q(\theta \mid h(x)), \theta) \right] + \lambda \cdot \mathbb{E}_{x^*\sim p^*(x)} \left[ C\left( \frac{p(x^* \mid \theta)\, p(\theta)}{q(\theta \mid h(x^*))} \right) \right], \quad (2)$$

where $S$ is a strictly proper score (Gneiting & Raftery, 2007) and $C$ is a self-consistency score. The neural networks to be optimized are a generative model $q$ and (optionally) a summary network $h$ extracting lower dimensional sufficient statistics from the data. We will call the first loss component, $\mathbb{E}_{(\theta,x)\sim p(\theta,x)} \left[ S(q(\theta \mid h(x)), \theta) \right]$, the (standard) simulation-based loss, as it forms the basis for standard ABI approaches using simulation-based learning. E.g., this is the maximum likelihood loss for normalizing flows (Kobyzev et al., 2020; Papamakarios et al., 2021) or a vector-field loss for flow matching (Liu et al., 2023; Lipman et al., 2023). We will refer to the second loss component as the (Bayesian) self-consistency loss.

In practice, we approximate the expectations in Eq. (2) with finite amounts of simulated and real training data. That is, for $N$ instances $(\theta_n, x_n) \sim p(\theta, x)$ and $M$ instances $x_m^* \sim p^*(x)$, we employ

$$(q^*, h^*) = \underset{q,h}{\mathrm{argmin}}\, \frac{1}{N} \sum_{n=1}^{N} \left[ S(q(\theta_n \mid h(x_n)), \theta_n) \right] + \lambda \cdot \frac{1}{M} \sum_{m=1}^{M} \left[ C\left( \frac{p(x_m^* \mid \theta_m)\, p(\theta_m)}{q(\theta_m \mid h(x_m^*))} \right) \right]. \quad (3)$$

where $\lambda$ is the self-consistency weight described in Appendix B. Asymptotically for $N \to \infty$, that is, for infinite training data generated from the simulator $p(\theta, x)$, a universal density estimator (Draxler et al., 2024) minimizing a strictly proper simulation-based loss (Gneiting & Raftery, 2007) is sufficient to ensure perfect posterior approximation for any data. By this, we mean that the posterior approximation becomes identical to the posterior we would obtain if we could analytically compute $p(\theta \mid x) = p(x \mid \theta)p(\theta)/p(x)$. This *analytic posterior* is sometimes also referred to as "true" or "correct" posterior (of the specified statistical model). In practice, the posterior is rarely analytic, but we can still verify the accuracy of an approximation by comparing it with the results of a gold-standard approach (if available), such as a sufficiently long, converged MCMC run (Magnusson et al., 2024).

While neural posterior approximation is perfect asymptotically, its pre-asymptotic performance, that is, when training $q(\theta \mid h(x))$ only on a finite amount of simulated data, can become arbitrarily bad (Frazier et al., 2024; Schmitt et al., 2023). For any data $x^*$ that is outside the data space implied by $p(\theta, x)$, for instance, when the model is misspecified (see Appendix B for definition), the posterior

approximation $q(\theta \mid h(x^*))$ may be arbitrarily far away from the analytic posterior $p(\theta \mid x^*)$. As a result, a simulation-based loss is insufficient to achieve robust ABI in practice. This is where the self-consistency loss comes in: As we will show, the latter greatly improves generalization to atypical data at inference time.

One particular choice for $C$ is the variance (with respect to $\theta \sim p_C(\theta)$) of the log Bayesian self-consistency ratio (Schmitt et al., 2024):

$$C \left( \frac{p(x^* \mid \theta) \, p(\theta)}{q(\theta \mid h(x^*))} \right) = \text{Var}_{\theta \sim p_C(\theta)} \left[ \log p(x^* \mid \theta) + \log p(\theta) - \log q(\theta \mid h(x^*)) \right], \tag{4}$$

which is motivated by theoretical results by Köthe (2023), who show that minimizing the variance of the Bayesian self-consistency ratio also minimizes the KL divergence between true and approximate posterior. $p_C(\theta)$ can be any proposal distribution over the parameter space. In practice, two natural choices for $p_C(\theta)$ are either the prior $p(\theta)$ or the current approximate posterior $q_t(\theta \mid h(x^*))$ at training iteration $t$. Because the largest contributions to $C$ are expected to be in high-density regions of the approximate posterior, we choose $q_t(\theta \mid h(x^*))$ for all of our experiments, see Appendix B for further details on practical considerations.

## 2.3 SELF-CONSISTENCY LOSSES ARE STRICTLY PROPER

Below, we discuss the strict properness of Bayesian self-consistency losses, showing that they can be used alongside simulation-based losses without altering the target posterior. To simplify the notation, we denote posterior approximators simply as $q(\theta \mid x)$ without considering architectural details such as the use of summary networks $h(x)$. All theoretical results and their proofs remain valid if $x$ is replaced by $h(x)$, provided that the summary network is expressive enough to learn sufficient statistics from $x$.

**Proposition 1.** *Let $C$ be a score that is globally minimized if and only if its functional argument is constant across the support of the posterior $p(\theta \mid x)$ almost everywhere. Then, $C$ applied to the Bayesian self-consistency ratio with known likelihood*

$$C \left( \frac{p(x \mid \theta) \, p(\theta)}{q(\theta \mid x)} \right) \tag{5}$$

*is a strictly proper loss: It is globally minimized if and only if $q(\theta \mid x) = p(\theta \mid x)$ almost everywhere.*

In particular, the variance loss (4) fulfills the assumptions of Proposition 1.

**Proposition 2.** *The loss (4) based on the variance of the log Bayesian self-consistency ratio is strictly proper if the support of $p_C(\theta)$ encompasses the support of $p(\theta \mid x)$.*

The proofs of Propositions 1 and 2 are provided in Appendix A. The strict properness extends to semi-supervised losses of the form (2), which combine standard simulation-based and self-consistency losses.

**Proposition 3.** *Under the assumptions of Proposition 1, the semi-supervised loss (2) is strictly proper for any choice of $p^*(x)$.*

Proposition 3 holds for every instantiation of $C$ that satisfies Proposition 1. The proof of Proposition 3 follows immediately from the fact that the sum of strictly proper losses is strictly proper. Importantly, since Proposition 3 holds independently of $p^*(x)$, it holds both in the case of a well-specified model, where $p^*(x) = p(x)$, and also in case of any model misspecification or domain shift where $p^*(x) \neq p(x)$. That is, there is *no trade-off* in the semi-supervised loss (2), since both loss components are globally minimized for the same target, that is, the analytic posterior.

In other words, adding the SC loss does not alter the underlying statistical model and continues to target the analytic posterior of that model. As far as we are aware, all other methods aimed at improving the robustness of ABI work by either explicitly or implicitly adjusting the statistical model or the target distribution (moving away from the analytic posterior), thus introducing some form of regularization. In contrast, SC losses should not be interpreted as regularizers: they are strictly proper losses that directly target the analytic posterior of the specified statistical model (i.e., Target 1 in the taxonomy of Elsemüller et al., 2025).

Our proposed semi-supervised loss combines standard simulation-based neural posterior estimation (NPE) and unsupervised self-consistency. Both target the analytic posterior, but in complementary ways: The standard NPE loss facilitates network training on simulated (labeled) data where ground-truth parameters are known. However, it does not ensure posterior accuracy on the empirical data distribution, particularly in regions where simulated data may be sparse or unrepresentative due to model misspecification. The self-consistency loss complements NPE by enforcing Bayes' rule on observed data and explicitly rewards correct marginal likelihood estimation. This ensures alignment between prior, posterior and likelihood on the empirical data distribution–a property not guaranteed by simulation-based losses alone.

For completeness, we also define strictly proper self-consistency losses for likelihood estimators, rather than posterior approximators.

**Proposition 4.** *Suppose the posterior $p(\theta \mid x)$ is known and the likelihood is estimated by $q(x \mid \theta)$. Then, under the assumptions of Proposition 1, Bayesian self-consistency ratio losses of the form*

$$C\left(\frac{q(x \mid \theta)\, p(\theta)}{p(\theta \mid x)}\right) \tag{6}$$

*are strictly proper: They are globally minimized if and only if $q(x \mid \theta) = p(x \mid \theta)$ almost everywhere.*

The proof of Proposition 4 proceeds in the same manner as for Proposition 1, just exchanging likelihood and posterior. Clearly, strict properness does not necessarily hold if *both* posterior and likelihood are unknown or approximate because any pair of approximators $q(\theta \mid x)$ and $q(x \mid \theta)$ that satisfy $q(\theta \mid x) \propto q(x \mid \theta)\, p(\theta)$ minimize the SC loss. For example, the choices $q(\theta \mid x) = p(\theta)$ and $q(x \mid \theta) \propto 1$ minimize the SC loss, but may be arbitrarily far from their actual target distributions $p(\theta \mid x)$ and $p(x \mid \theta)$, respectively.

In other words, if both likelihood and posterior are unknown, the SC loss has to be coupled with another loss component, such as the maximum likelihood loss, to enable joint learning of both approximators $q(\theta \mid x)$ and $q(x \mid \theta)$. Nevertheless, SC still yields notable improvements: in our experiments, the semi-supervised loss (2) considerably enhanced the robustness of ABI *even when both the posterior and likelihood are unknown*.

## 3 RELATED WORK

The robustness of ABI and simulation-based inference methods more generally has been the focus of multiple recent studies (e.g., Frazier et al., 2020; Frazier & Drovandi, 2021; Frazier et al., 2024; Dellaporta et al., 2022; Ward et al., 2022; Cemgil et al., 2020; Gloeckler et al., 2023; Huang et al., 2023; Gao et al., 2023; Siahkoohi et al., 2023; Kelly et al., 2024; Wehenkel et al., 2024; Pacchiardi et al., 2024; Schmitt et al., 2023). These efforts can be broadly classified into two categories: (a) analyzing or detecting simulation gaps and (b) mitigating the impact of simulation gaps on posterior estimates. Since our work falls into the latter category, we briefly discuss methods aimed at increasing the robustness of fully amortized approaches. E.g., Gloeckler et al. (2023) explore efficient regularization techniques that trade off some posterior accuracy to enhance the robustness of posterior estimators against adversarial attacks. Ward et al. (2022) and Siahkoohi et al. (2023) apply *post hoc* corrections based on real data, utilizing MCMC and the reverse KL-divergence, respectively.

Differently, Gao et al. (2023) propose a departure from standard Bayesian inference by minimizing the expected distance between simulations and observed data, akin to generalized Bayesian inference with scoring rules (Pacchiardi et al., 2024). Perhaps the closest work in spirit to ours is Wehenkel et al. (2024), which introduces the use of additional training information in the form of a (labeled) calibration set $(x^*, \theta^*)$ that contains observables from the real data distribution as well as the corresponding ground-truth parameters.

In contrast to the methods above, our approach (a) avoids trade-offs between accuracy and robustness, (b) requires no modifications to the neural estimator after training, therefore fully maintaining inference speed, (c) affords proper Bayesian inference, and (d) does not assume known ground truth parameters for a calibration set. Thus, it can be viewed as one of the first instantiations of *semi-supervised* ABI.

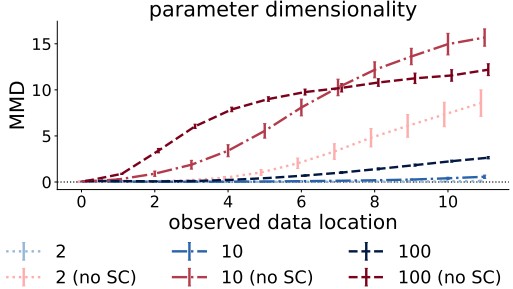 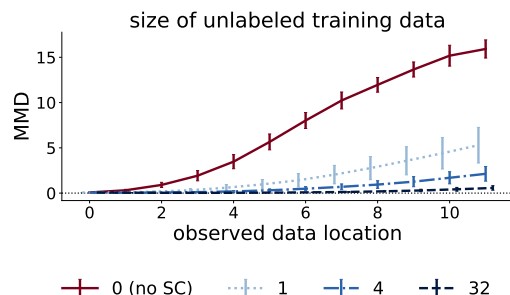

**(a):** Posterior distance between approximate and true posterior for varying parameter dimensionality.

**(b):** Posterior distance between approximate and true posterior for varying unlabeled training data sizes.

Figure 2: Posterior distance quantified by maximum mean discrepancy (MMD) to the analytic posterior for variations of the default configuration. Errorbars show $\pm 1$ SDs over 10 model refits.

## 4 CASE STUDIES

### 4.1 MULTIVARIATE NORMAL MODEL

We first illustrate the usefulness of our proposed self-consistency loss on a controllable toy problem (Schmitt et al., 2023). The prior and likelihood are given by

$$\theta \sim \text{Normal}(\mu_{\text{prior}}, \sigma_{\text{prior}}^2 I_D), \qquad x^{(k)} \sim \text{Normal}(\theta, \sigma_{\text{lik}}^2 I_D) \tag{7}$$

The parameters $\theta \in \mathbb{R}^D$ are sampled from a $D$-dimensional multivariate normal distribution with mean vector $\mu_{\text{prior}}$ and diagonal covariance matrix $\sigma_{\text{prior}}^2 I_D$. Here, we fix $\mu_{\text{prior}} = 0$ and $\sigma_{\text{prior}}^2 = 1$. On this basis, $K$ independent, synthetic data points $x^{(k)} \in \mathbb{R}^D$ are sampled from a $D$-dimensional multivariate normal distribution with mean vector $\theta$ and diagonal covariance matrix $\sigma_{\text{lik}}^2 I_D$. We set $\sigma_{\text{lik}}^2 = K$ such that the total information in $x$ remains constant across values of $K$, which simplifies comparisons across observations of varying numbers of data points. Refer to Appendix C for more details on training and neural architectures.

In our numerical experiments, we study the influence of several aspects of the normal model on the performance of NPE. To prevent combinatorial explosion, we vary the factors below separately, with all other factors fixed to their default configuration (highlighted in **bold**): (1) parameter dimensionality ($D = 2, \mathbf{10}, 100$), (2) number of unlabeled observations for the self-consistency loss $\{x_m^*\}_{m=1}^M$ ($M = 1, 4, \mathbf{32}$), (3) mean $\mu^*$ of the unlabeled observations $x_m^*$ ($\mu^* = 0, 1, 2, \mathbf{3}, 5$), (4) inclusion of a summary network ($K = 10$) or **not** ($\mathbf{K = 1}$), (5) likelihood function (**known**, estimated).

**Results** In Figure 1, we depict the results obtained from (a) standard NPE (trained on the simulation-based loss only), (b) our semi-supervised NPE + SC (with self-consistency loss on known likelihood), and (c) the gold-standard (analytic) reference. We see that standard NPE already completely fails for $x_{\text{obs}} \sim N(\mu_{\text{obs}} = 2, 0.01 I_D)$, and subsequently also for any larger values $\mu_{\text{obs}} > 2$. In contrast, our semi-supervised approach achieves almost perfect posterior estimation. This holds true even in cases where $x_{\text{obs}}$ is multiple standard deviations away from *all* the training data, that is, from both the labeled dataset $\{(\theta_n, x_n)\}_{n=1}^N$ and the unlabeled dataset $\{x_m^*\}_{m=1}^M$. These results indicate that the SC criterion can provide strong robustness gains even far outside the typical space of training data.

In Figure 2, we report the maximum mean discrepancy (MMD) between the approximate and true posterior the factors parameter dimensionality and number of unlabeled observations. When varying the parameter dimensionality (Figure 2a), including the SC loss yields nearly perfect posterior approximation up to 10 dimensions, even with extreme deviations from the initial training data. It also significantly improves accuracy in the 100 parameter scenario. The dataset size factor (Figure 2b) shows robust gains, with clear improvements over NPE even when using as few as four unlabeled observations (versus 1024 labeled ones). In Figure 6 (Appendix D), we report posterior mean and standard deviation bias as well as maximum mean discrepancy for all the above factors. Varying the mean $\mu_{\text{obs}}$ of the new observations shows that, as long as the data used for evaluating the SC loss is

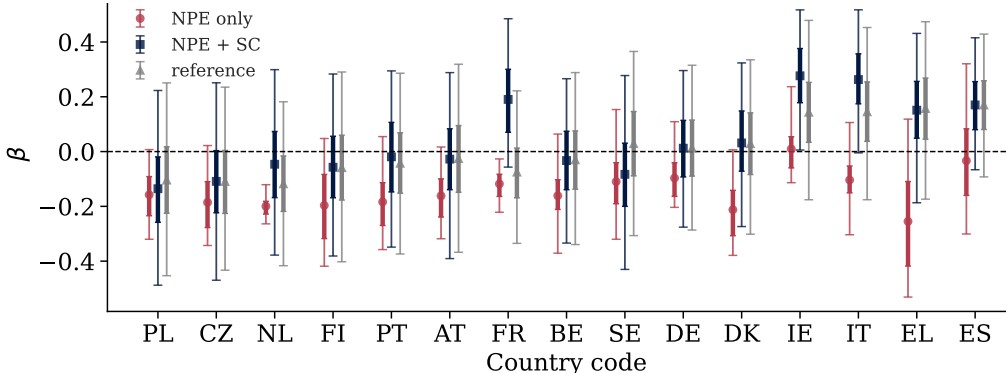

Figure 3: Comparison of posterior estimates for 15 countries (ISO 3166 alpha-2 codes) among standard NPE (red), NPE + SC (blue), and Stan (reference in gray; Carpenter et al., 2017). Central 50% (thick lines) and 95% (thin lines) posterior intervals of the autoregressive component $\beta$ are shown, sorted by lower 5% quantile as per Stan (i.e., established benchmark). The SC loss was evaluated on data from $M = \mathbf{8}$ countries during training, greatly enhancing ABI's robustness in both no-misspecification scenarios and real-data evaluations.

not identical to the training data (i.e., as long as $\mu_{\text{obs}} \neq 0$), including the SC loss component enables accurate posterior approximation far outside the typical space of the training data.

In Figure 7 in Appendix D, we see that the benefits of self-consistency persist when the posterior is conditioned on more than one data point per observation ($K = 10$), that is, in the presence of a summary network. Further, we still see clear benefits of adding the self-consistency loss even when the likelihood is estimated by a neural likelihood approximator $q(x \mid \theta)$, trained jointly with the posterior approximator $q(\theta \mid x)$ on the same training data. However, with an estimated likelihood, posterior bias, especially bias in the posterior standard deviation, and MMD distance to the true posterior are larger than in the known likelihood case. We also compared our semi-supervised approach with other robustness methods - NPE-DANN, NPE-MMD, and NNPE. The results described in Appendix D.1 demonstrate the benefits of our approach over other methods.

## 4.2 FORECASTING AIR PASSENGER TRAFFIC: AN AUTOREGRESSIVE MODEL WITH PREDICTORS

We apply our semi-supervised loss to analyze trends in European air passenger traffic data provided by Eurostat (2022a;b;c). This case study highlights that the strong robustness gains also occur in real-world scenarios and model classes that are challenging to estimate in a simulation-based inference setting. We retrieved time series of annual air passenger counts between 15 European countries (departures) and the USA (destination) from 2004 to 2019 and fit the following autoregressive process of order 1:

$$y_{j,t+1} \sim \text{Normal}(\alpha_j + y_{j,t}\beta_j + u_{j,t}\gamma_j + w_{j,t}\delta_j, \sigma_j), \tag{8}$$

where the target quantity $y_{j,t+1}$ is the difference in air passenger traffic for country $j$ between time $t + 1$ and $t$. To predict $y_{j,t+1}$ we use two additional predictors: $u_{j,t}$ is the annual household debt of country $j$ at time $t$, measured in % of gross domestic product (GDP) and $w_{j,t}$ is the real GDP per capita. Training relies on a small simulation budget of $N = 1024$, with the self-consistency loss evaluated on real data from $M \in \{4, 8, 15\}$ countries. Further details on model and training are provided in Appendix E.

**Results**  In Figure 3, we show exemplary results from standard NPE, our semi-supervised NPE + SC ($M = 8$), and Stan (Carpenter et al., 2017) as reference. We see that NPE is highly inaccurate for many countries, whereas NPE + SC is in strong agreement with the reference for all but one country. As shown in Table 1, adding the self-consistency loss ($M = 8$) strongly improves posterior estimates for all five parameters across all metrics, on average across countries. The complete results can be found in Appendix F.

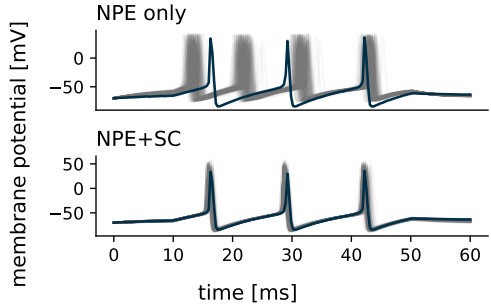 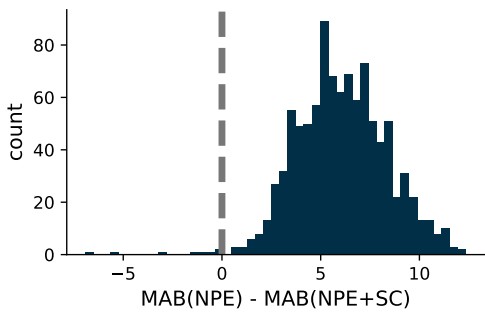

**(a):** Posterior predictive samples without (top row) and with (bottom row) self-consistency loss.

**(b):** Quantitative evaluation of predictive bias.

Figure 4: (a) Posterior predictive samples (gray) inferred from an out-of-simulation dataset (black). NPE only produces highly biased predictions while NPE+SC yields accurate results. (b) Histogram of the mean absolute bias (MAB) difference of posterior predictions computed for 1000 out-of-simulation datasets. NPE+SC has lower bias than NPE for almost all datasets.

## 4.3 HODGKIN-HUXLEY MODEL OF NEURON ACTIVATION

To investigate the effect of the self-consistency loss on a model involving high-dimensional data, we evaluate our approach on the Hodgkin-Huxley model, which was previously used in an ABI setting by Gloeckler et al. (2023). The Hodgkin-Huxley model is a classical model in neuroscience to describe neuron activation via a set of 5 ordinary differential equations. In brief, the model has 7 parameters (electrical conductances of different ion channels, membrane capacitance and reversal potentials), and the output $y_i$ with observation index $i$ is a 200-dimensional time series of the membrane potential. A full definition of the model, as well as the training setup and a description of the neural architectures, are provided in Appendix G.

To facilitate network training, all parameters are transformed to follow standard normal distributions through appropriate transformations. For example, the parameter $g_{\text{Na}}$ with marginal prior $g_{\text{Na}} \sim \text{LogNormal}(\log(110), 0.1)$ is transformed via $z_{g_{\text{Na}}} = (\log(g_{\text{Na}}) - \log(110))/0.1$. We denote the full set of transformed model parameters by $\theta$. Training is performed with a simulation budget of $N = 32{,}768$. For each loss evaluation, the self-consistency loss is computed on a random subset of 32 samples drawn from a pool of $M = 1{,}024$ unlabeled observations. These are generated by first sampling $\theta \sim \text{Normal}(0, 2)$, applying the inverse transformations to recover the original parameter scale, and then simulating time series of the membrane potential as above.

**Results** To assess the benefits of our approach in the out-of-distribution setting, Figure 4a shows posterior predictive samples inferred from data simulated with $\theta \sim \text{Normal}(-2, 1)$. This contrasts with training data from $\theta \sim \text{Normal}(0, 1)$ and self-consistency evaluation data from $\theta \sim \text{Normal}(0, 2)$. When training without the self-consistency loss, the neural posterior density estimator produces samples inconsistent with the observed data, while incorporating the loss yields accurate predictions (see Appendix H for further results). To quantify this, Figure 4b reports mean absolute bias differences between the two estimators. The self-consistency loss consistently and strongly improves predictions across the majority of time series. Even in the worst cases, it is at least competitive with the estimator trained without the self-consistency loss.

## 4.4 BAYESIAN DENOISING OF MNIST IMAGES

Finally, we demonstrate the utility of our self-consistency loss based approach in a high-dimensional (for ABI) image denoising task, following a set-up similar to Elsemüller et al. (2025). The parameter vector $\theta \in \mathbb{R}^{784}$ is the flattened image, and the observation $x \in \mathbb{R}^{784}$ is a blurred version generated by a simulated noisy camera. We assume an implicit prior $\theta \sim p(\theta)$ defined by a generative model trained on blurred MNIST images of the digit "0" (LeCun et al., 1998), and an implicit likelihood

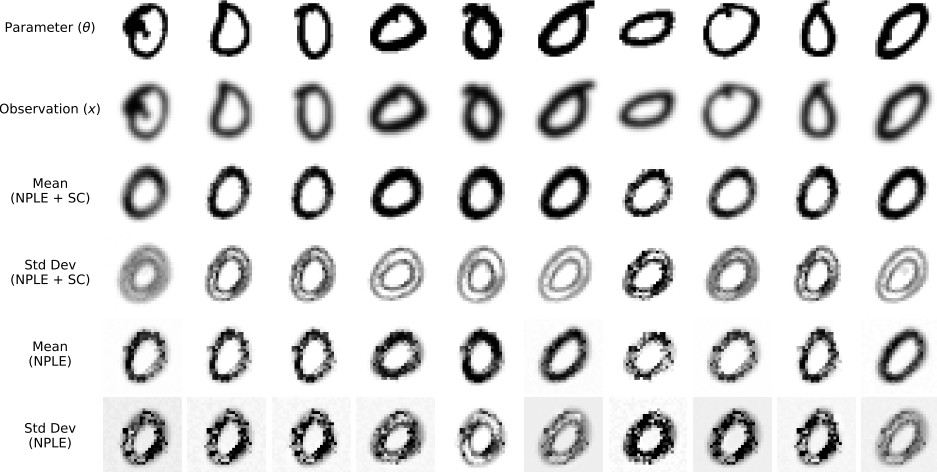

Figure 5: Example of denoising results for MNIST images of digit "0" in the held-out test set. The *first row* shows ten randomly selected MNIST images ($\theta$), the *second row* depicts images after applying the Gaussian blur ($x$), *third* and *fourth rows* depict mean and standard deviation (SD) of 500 posterior samples estimated from the corresponding blurry observations using NPLE+SC, and the *fifth* and *sixth rows* depict the mean and SD of 500 posterior samples using NPLE only. Incorporating SC loss significantly improves denoising.

$p(x|\theta)$ implemented by reapplying the same blur to each $\theta$. This creates a challenging neural posterior-likelihood estimation (NPLE) problem.

To construct the training set, we blur all digit "0" images in the MNIST training set with a fixed Gaussian filter and train a neural network to (i) sample blurred simulated images $\{\theta^i\}_{i=1}^N \sim p(\theta)$ and (ii) evaluate their log-probabilities. For each sample $\theta^i$, we then generate an observation $x^i \sim p(x|\theta^i)$ by reapplying the same Gaussian blur, yielding $N = 12000$ pairs $(\theta^i, x^i)$ to train posterior and likelihood networks. For the SC loss, we use 400 held-out MNIST test images blurred only by the likelihood model, (i.e., no additional prior blur), inducing deliberate prior misspecification and enabling us to asses the robustness of SC in NPLE. More details about the training set-up and model architecture can be found in the Appendix I.

**Results**   We perform inference on a separate subset of 580 MNIST test images. Figure 5 and Figure 16 in Appendix J show randomly chosen examples alongside the posterior mean and standard deviation maps computed from 500 posterior samples. Reconstructions using the self-consistency approach (NPLE+SC) are smoother and more faithful to the ground-truth, whereas NPLE reconstructions are pixelated and blurry. Moreover, NPLE+SC yields coherent uncertainty maps, with elevated variance localized along digit contours where edge ambiguity is expected. In contrast, NPLE estimates exhibit scattered, patchy uncertainty across the digit and background. Additional examples are reported in Appendix J.

## 5   DISCUSSION

We demonstrated that Bayesian self-consistency losses significantly increase the robustness of neural amortized Bayesian inference (ABI) on out-of-simulation data. Accurate inference outside the training distribution, such as in the presence of model misspecification, has long posed a major challenge for ABI. While self-consistency was originally introduced to improve training efficiency with slow simulators (Schmitt et al., 2024; Ivanova et al., 2024), it had not been previously explored as a remedy to simulation gaps. Existing supervised ABI approaches have been known to dramatically fail in such cases (Schmitt et al., 2023; Gloeckler et al., 2023; Huang et al., 2023), as we also illustrated in our experiments. In contrast, when optimizing for self-consistency on *unlabeled* out-of-simulation data, we obtained nearly unbiased posterior estimation far beyond the training distribution. The strong robustness gains persisted even in models with several hundred parameters. However, using an

approximate (neural) likelihood has not yet matched the robustness of the known-likelihood case. Finally, as self-consistency losses do not require data labels (i.e., true parameter values), we can use any amount of *real data* during training to improve the robustness of ABI.

**Limitations and future directions**   Our variance-based self-consistency loss relies on fast density evaluations during training, keeping times competitive. This makes free-form methods such as flow matching (Lipman et al., 2023) or score-based diffusion (Song et al., 2020) less practical due to their need for numerical integration. As a result, efficient self-consistency losses for free-form flows, along with joint learning of posteriors and very high-dimensional likelihoods, remains an open avenue for future research.

Our semi-supervised approach can also be combined with post-hoc correction methods such as Pareto-smoothed importance sampling (Vehtari et al., 2024). When applied to NPE alone, importance sampling often fails because the learned posterior can deviate substantially from the analytic target (Li et al., 2024). By contrast, our approach produces posterior approximations that are much closer to the analytic posterior, making subsequent importance sampling more effective. This combination therefore has the potential to substantially expand the range of scenarios in which accurate estimation of the analytic posterior is feasible.

## Acknowledgments

Daniel Habermann and Paul Bürkner acknowledge support of the Deutsche Forschungsgemeinschaft (DFG, German Research Foundation) Projects 508399956 and 528702768. Paul Bürkner further acknowledges support of the DFG Collaborative Research Center 391 (Spatio-Temporal Statistics for the Transition of Energy and Transport) – 520388526. Stefan T. Radev is supported by NSF under Grant No. 2448380. Aayush Mishra acknowledges the computing time provided on the Linux HPC cluster at Technical University Dortmund (LiDO3).

## Reproducibility Statement

We provide all details necessary to reproduce our results. Hyperparameter settings and network architectures are described in Section 4 and Appendices C, D, E, G, and I. The code to run all experiments is provided in the abstract. Hardware specifications are reported in Appendix K. All theoretical results are accompanied by complete proofs in Section 2 and Appendix A.

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

APPENDIX

## A  PROOFS

*Proof of Proposition 1.* By assumption, $C$ is globally minimized if and only if

$$\frac{p(x \mid \theta) \, p(\theta)}{q(\theta \mid x)} = A \tag{9}$$

for some constant $A$ (independent of $\theta$) almost everywhere over the posterior's support. Accordingly, any approximate posterior solution $q(\theta \mid x)$ that attains this global minimum has to be of the form

$$q(\theta \mid x) = p(x \mid \theta) \, p(\theta) \, / \, A. \tag{10}$$

By construction, $q(\theta \mid x)$ is a proper probability density function, so it integrates to 1. It follows that

$$1 = \int q(\theta \mid x) \, d\theta = \int p(x \mid \theta) \, p(\theta) \, d\theta \, / \, A = p(x) \, / \, A. \tag{11}$$

Rearranging the equation yields $A = p(x)$ and thus

$$q(\theta \mid x) = p(x \mid \theta) \, p(\theta)/p(x) = p(\theta \mid x) \tag{12}$$

almost everywhere. □

*Proof of Proposition 2.* The variance over a distribution $p_C(\theta)$ reaches its global minimum (i.e., zero), if and only if its argument is constant across the support of $p_C(\theta)$. Because the log is a strictly monotonic transform,

$$\log p(x^* \mid \theta) + \log p(\theta) - \log q(\theta \mid x^*) = \log A \tag{13}$$

for some constant $A$ implies

$$\frac{p(x \mid \theta) \, p(\theta)}{q(\theta \mid x)} = A, \tag{14}$$

which is sufficient to satisfy the assumptions of Proposition 1. □

## B  FREQUENTLY ASKED QUESTIONS

**Q: How do you define model-misspecification, simulation-gap, and domain shift?**

We define **model misspecification** as a mismatch between an unknown (true) distribution of data $x^* \sim p^*(x)$ coming from a complex, real-world system, and the marginal likelihood of data $p(x|\mathcal{M})$ under some model $\mathcal{M}$. Concretely, we call a model misspecified if

$$p^*(x) \neq p(x \mid \mathcal{M}) \coloneqq \int_{\Theta} p(x \mid \theta, \mathcal{M}) \, p(\theta \mid \mathcal{M}) \, d\theta, \quad \forall x \in \mathcal{X}.$$

Within this framework, a **simulation gap** refers to the divergence between the empirical distribution of observed data $p^*(x)$ and the empirical distribution of simulated data $\hat{p}(x)$, where $\hat{p}(x)$ is obtained via finite samples $x \sim p(x \mid \theta, \mathcal{M})$ for $\theta \sim p(\theta \mid \mathcal{M})$. That is, even when the model is not misspecified, limitations in simulation coverage can create gaps in support.

A **domain shift** refers to any distributional discrepancy between the simulated training data ($x \sim p(x \mid \mathcal{M})$) and the observed data ($x^* \sim p^*(x)$). This includes, but is not limited to, model misspecification and simulation gaps, and encompasses broader generalization challenges faced when deploying amortized inference methods outside the simulation domain.

**Q: What is the intuition behind using self-consistency to make the inference robust?**

In a well-specified setting, the simulator accurately captures the true data-generating process and the learned posterior matches the true posterior. In this case, neural networks only suffer from approximation error due to finite data or model capacity. Under model misspecification, the observed

data are drawn from a distribution $p^*(x) \neq p(x)$, and hence the model-implied posterior $p(\theta \mid x) \propto p(x \mid \theta)p(\theta)$ does not correspond to the posterior under $p^*(x)$ but under $p(x)$.

In such settings, neural networks face two types of error: approximation error, as before, due to imperfect learning, and extrapolation error, because the network encounters observations $x \sim p^*(x)$ that may lie outside the simulation support, and thus lacks guidance on how to sample from the posterior in those regions. In this case, minimizing the self-consistency loss encourages the learned posterior $q(\theta \mid x)$ to remain internally consistent with the assumed model $p(x, \theta)$ when evaluated on empirical samples $x \sim p^*(x)$. It enforces that the learned posterior $q(\theta \mid x)$ remains consistent with the model assumptions, effectively guiding the network to extrapolate in a model-coherent way. That is, even when the data are out-of-distribution relative to the simulator, the network learns to sample in a way that is self-consistent under the (misspecified) model posterior, improving robustness despite model mismatch.

**Q: Why is the self-consistency loss coupled with standard simulation-based loss when self-consistency is strictly proper?**

In principle, minimizing the self-consistency loss is sufficient to recover the true posterior, as guaranteed by Proposition 1. However, in practice, we observe that using the SC loss as the sole training objective leads to initialization and convergence issues. This is primarily due to the density-based nature of the SC loss. At initialization, when the approximate posterior and/or likelihood are poorly calibrated, the self-consistency loss can take on extreme values, often resulting in numerical instabilities, such as exploding gradients. Consequently, optimization tends to diverge or fail to make meaningful progress. We observe that the SC loss in tandem with the NPE loss provides a more stable and effective learning signal. The amortized loss anchors the model early on by using supervised simulated data, while the SC loss complements it by enforcing consistency on observed data, especially in out-of-simulation regimes.

**Q: What is the motivation behind using the variance over the log Bayesian self-consistency ratio for $C$?**

The choice of using the variance of the logarithm of the Bayesian self-consistency ratio, evaluated over $p \sim p_C(\theta)$ (Eq. 4), is motivated as follows: Köthe (2023) (Eq. 33) shows that a second-order Taylor expansion of the regular NPE loss function yields $KL(p(\theta \mid x) || q(\theta \mid x)) \approx \frac{1}{2p(x)^2} \text{Var}_{p(\theta|x)} \left( \frac{p(\theta)p(x|\theta)}{q(\theta|x)} \right)$. From this, it follows that minimizing $\text{Var}(\dots)$ also minimizes the KL divergence between true and approximate posterior. Because the denominator in $\text{Var}(\dots)$ can cause numerical instabilities due to vanishingly small values, we instead target $\text{Var}(\log(\dots))$. Minimizing this quantity via a loss function still minimizes the KL divergence between the true and approximate posterior, which follows from Proposition 1.

**Q: What is the motivation behind using the approximate posterior $q_t(\theta \mid h(x^*))$ at training iteration $t$ for $p_C(\theta)$?**

Rearranging Bayes' rule shows that the marginal likelihood $p(x) = \frac{p(\theta)p(x|\theta)}{p(\theta|x)}$ is constant for all $\theta$. If we replace the true posterior $p(\theta \mid x)$ by an approximate posterior $q(\theta \mid x)$, large fluctuations in the marginal likelihood landscape are expected to be in high-density regions of the approximate posterior. We therefore choose the approximate posterior as the proposal distribution.

The choice of using the variance of the logarithm of the Bayesian self-consistency ratio, evaluated over $p \sim p_C(\theta)$ (Eq. 4), is motivated as follows: Köthe (2023) (Eq. 33) shows that a second-order Taylor expansion of the regular NPE loss function yields $KL(p(\theta \mid x) || q(\theta \mid x)) \approx \frac{1}{2p(x)^2} \text{Var}_{p(\theta|x)} \left( \frac{p(\theta)p(x|\theta)}{q(\theta|x)} \right)$. From this, it follows that minimizing $\text{Var}(\dots)$ also minimizes the KL divergence between true and approximate posterior. Because the denominator in $\text{Var}(\dots)$ can cause numerical instabilities due to vanishingly small values, we instead target $\text{Var}(\log(\dots))$. Minimizing this quantity via a loss function still minimizes the KL divergence between the true and approximate posterior, which follows from Proposition 1.

**Q: How to choose the self-consistency weight $\lambda$?**

From the strict-properness of self-consistency losses, it follows that $\lambda$ does not control a trade-off between the standard NPE loss and the SC loss, as both optimize for the same training objective. Instead, its main role is to stabilize training by preventing initialization and convergence issues. For

experiments with low-dimensional parameter spaces, a default choice of $\lambda = 1$ typically works well. However in high-dimensional settings, such as the image-denoising experiment, the approximate posterior can be particularly bad in the early epochs, leading to unstable training. In such cases, we found it to be effective to use a linear schedule for $\lambda$ that keeps it at 0 for the first few epochs and then gradually increases to 1.

## C  DETAILED SETUP OF THE MULTIVARIATE NORMAL CASE STUDY

From the multivariate normal model described in Section 4.1, we simulate a *labeled* training dataset with a budget of $N = 1024$, that is, $N$ independent instances of $\theta_n$ (the "labels") with corresponding observations $x_n = \{x_n^{(k)}\}_{k=1}^K$, each consisting of $K$ data points. This labeled training dataset $\{(\theta_n, x_n)\}_{n=1}^N$ is used for optimizing the standard simulation-based loss component. The self-consistency loss component is optimized on an additional *unlabeled* dataset $\{x_m^*\}_{m=1}^M$ of $M = 32$ independent sequences $x_m^* = \{x_m^{*(k)}\}_{k=1}^K$, which, for the purpose of this case study, are simulated from

$$x_m^{*(k)} \sim \text{Normal}(\mu^*, I_D). \tag{15}$$

Since the self-consistency loss does not need labels (i.e., the true parameters having generated $x_m^*$), we could have also chosen any other source for $x^*$, for example, real-world data. Within each training iteration $t$, the variance term within the self-consistency loss was computed from $L = 32$ samples $\theta^{(l)} \sim q_t(\theta \mid x_m^*)$ from the current posterior approximation.

To evaluate the accuracy and robustness of the NPEs, we perform posterior inference on completely new observations $x_{\text{obs}} = \{x_{\text{obs}}^{(k)}\}_{k=1}^K$ , each consisting of $K$ independent data points sampled from

$$x_{\text{obs}}^{(k)} \sim \text{Normal}(\mu_{\text{obs}}, \sigma_{\text{obs}}^2 = 0.01 I_D). \tag{16}$$

The mean values $\mu_{\text{obs}} \in \{0, 1, \ldots, 11\}$ are progressively farther away from the training data. While conceptually simple and synthetic, this setting is already extremely challenging for simulation-based inference algorithms because of the large simulation gap (Schmitt et al., 2023): standard NPEs are only trained on (labeled) training data that are several standard deviations away from the observed data the model sees at inference time.

The faithfulness of the approximated posteriors $q(\theta \mid x_{\text{obs}})$ are assessed by computing the bias in posterior mean and standard deviation as well as the maximum mean discrepancy (MMD) with a Gaussian kernel (Gretton et al., 2012) between the approximate and true (analytic) posterior.

The analytic posterior for the normal means problem is a conjugate normal distribution

$$p(\theta \mid x_{\text{obs}}) = \text{Normal}(\mu_{\text{post}}, \sigma_{\text{post}}^2 I_D), \tag{17}$$

where $\mu_{\text{post}}$ is a $D$-dimensional posterior mean vector with elements

$$(\mu_{\text{post}})_d = \sigma_{\text{post}}^2 \left( \frac{\mu_{\text{prior}}}{\sigma_{\text{prior}}^2} + \frac{K(\bar{x}_{\text{obs}})_d}{\sigma_{\text{lik}}^2} \right), \tag{18}$$

$\sigma_{\text{post}}^2$ is the posterior variance (constant across dimensions) given by

$$\sigma_{\text{post}}^2 = \left( \frac{1}{\sigma_{\text{prior}}^2} + \frac{K}{\sigma_{\text{lik}}^2} \right)^{-1}, \tag{19}$$

and $(\bar{x}_{\text{obs}})_d$ is the mean over the $D$th dimension of the $K$ new data points $\{x_{\text{obs}}^{(k)}\}_{k=1}^K$.

For the NPEs $q(\theta \mid x)$, we use a neural spline flow (Durkan et al., 2019) with 5 coupling layers of 128 units each utilizing ReLU activation functions, L2 weight regularization with factor $\gamma = 10^{-3}$, 5% dropout and a multivariate unit Gaussian latent space. The network is trained using the Adam optimizer for 100 epochs with a batch size of 32 and a learning rate of $5 \times 10^{-4}$. These settings were the same for both the standard simulation-based loss and our proposed semi-supervised loss. For the conditions involving an estimated likelihood $q(x \mid \theta)$, we use the same configuration for the likelihood network as for the posterior network. For the summary network $h(x)$ (if included), we

use a deep set architecture (Zaheer et al., 2017) with 30 summary dimensions and mean pooling, 2 equivariant layers each consisting of 2 dense layers with 64 units and a ReLU activation function. The inner and outer pooling functions also use 2 dense layers with the same configuration. The likelihood network as well as the summary network are jointly trained with the inference network using the Adam optimizer for 100 epochs with a batch size of 32 and a learning rate of $5 \times 10^{-4}$.

## D  COMPREHENSIVE RESULTS FOR THE MULTIVARIATE NORMAL CASE STUDY

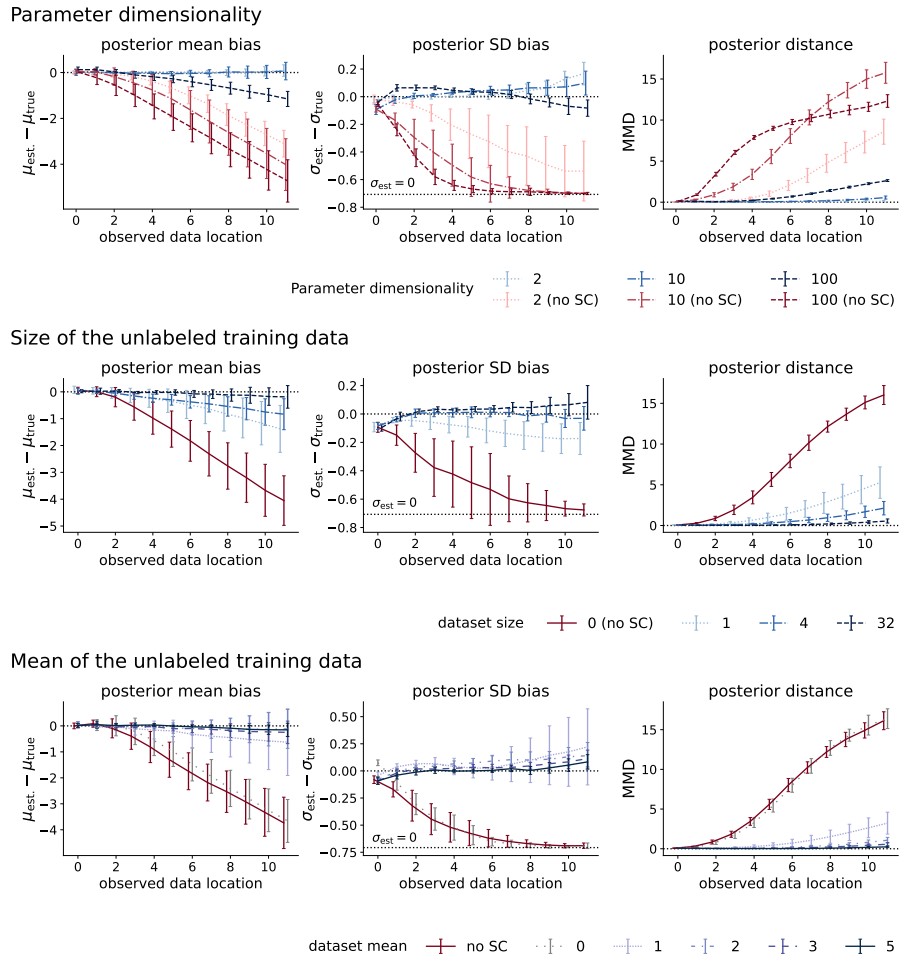

Figure 6: Bias of posterior mean, bias of posterior standard deviation and posterior distance quantified by maximum mean discrepancy to the analytic posterior for variations of the default configuration outlined in Section 4.1. NPE approximators with the added self-consistency loss component are shown in blue, NPE approximators using just the standard simulation-based loss are shown in red. Irrespective of the varied factor and for all metrics, adding the self-consistency loss component is always a drastic improvement over the standard simulation-based loss alone. The plots show that adding the self-consistency loss component provides strong robustness gains even in high-dimensional spaces (top row) or when the self-consistency loss is evaluated on little data (center row). Variation of the mean of the unlabeled training data show that adding the self-consistency loss drastically improves posterior estimation as long as data used for evaluating the self-consistency loss is at least slightly out-of-distribution compared to the original training data ($\mu^* \geq 1$). Errorbars show $\pm 1$ standard deviations over 10 model refits on new training data.

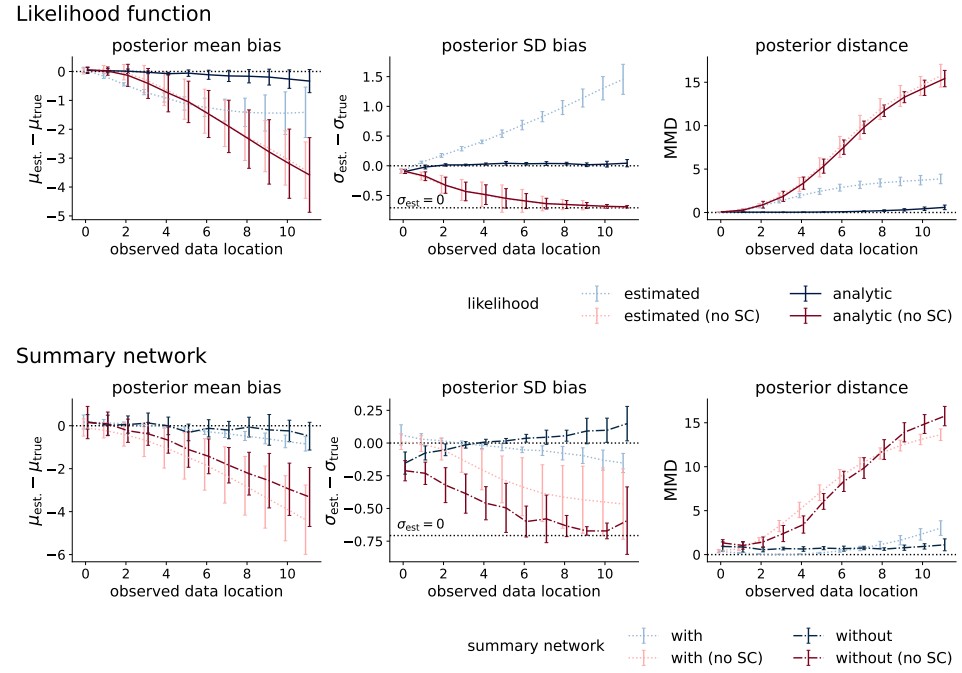

Figure 7: Bias of posterior mean, bias of posterior standard deviation and posterior distance quantified by maximum mean discrepancy to the analytic posterior when the likelihood is estimated (top row) and in presence of a summary network ($K = 10$ data points; bottom row). In the setting where the likelihood function is estimated, we observe a lower bias of the posterior mean and lower maximum mean discrepancy to the true posterior when the self-consistency loss component is added compared to the standard simulation-based loss alone. However, we do see some bias of the posterior standard deviation, although with reversed signed compared to the standard loss. The self-consistency loss provides strong robustness gains in the presence of a summary network (and known likelihood) in terms of all metrics. Errorbars show $\pm 1$ standard deviations over 10 model refits on new training data.

### D.1 COMPARISON TO OTHER ROBUSTNESS METHODS

Existing approaches for enhancing the robustness of amortized Bayesian inference typically fall into one of three categories, each with its own limitations. First, regularization-based methods (e.g., Gloeckler et al., 2023; Huang et al., 2023) introduce additional constraints inevitably altering the original training objective, thereby deviating from the true Bayesian posterior that the self-consistency loss seeks to approximate.

Second, some methods employ supervised corrections using a small amount of real-world, labeled data (e.g., Wehenkel et al., 2024) which requires access to ground-truth parameters, shifting the setting from semi-supervised to fully supervised learning.

Third, unsupervised domain adaptation (UDA) techniques such as NPE-DANN and NPE-MMD have been proposed to improve robustness under domain shift by aligning the distributions of simulated and observed data in the summary feature space (Elsemüller et al., 2025). These losses govern the alignment of the summary space between simulated and observed data, effectively adjusting the observed data resulting in posteriors conditioned on a transformed version of the observation (Target 2, Section 3, Elsemüller et al., 2025), rather than the analytic posterior. Moreover, as shown by Elsemüller et al. (2025), these UDA methods and NNPE (Ward et al., 2022) exhibit notably poor performance under prior misspecification.

That said, we conducted additional experiments with NPE-MMD (Huang et al., 2023), NPE-DANN (Elsemüller et al., 2025), and NNPE (Ward et al., 2022) on the two-dimensional multivariate normal model, replicating the prior misspecification setting described in Section 4.1. Since both UDA

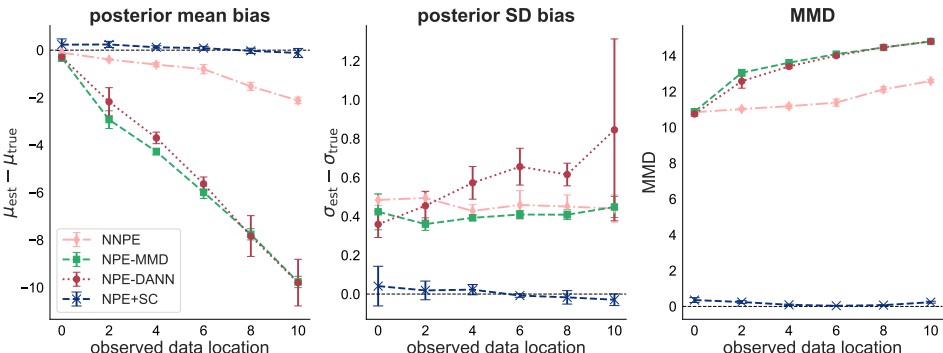

Figure 8: Bias of posterior mean, bias of standard deviation, and posterior distance quantified by maximum mean discrepancy, relative to the analytic posterior. NPE + SC provides far more robustness gains compared to NPE-MMD, NPE-DANN, and NNPE for the case of prior misspecification. Errorbars show $\pm 1$ standard deviations over 10 model refits on new training data.

methods rely on a summary network for feature alignment, we employed a shared DeepSet architecture (2-layer network with 30-dimensional output) across all methods, including NPE+SC, to ensure a fair comparison. For self-consistency loss, we used 32 unlabeled observed data at $\mu_{obs} = 3$ during training. We used Adam optimizer with a learning rate of 0.0005 trained for 40 epochs using 1,028 simulations. All the hyperparameter configurations were kept the same as described in Section 4.1 and Appendix C. For NPE-MMD, NPE-DANN, and NNPE, we utilized the implementation by Elsemüller et al. (2025).

The results clearly show that NPE-MMD, NPE-DANN, and NNPE fail under prior misspecification. Also, with increase in misspecification, the results for these methods became even worse while NPE+SC maintains the robustness gains. UDA methods also require large unlabeled data during training while we see strong robustness gains in NPE+SC with as few as 4 unlabeled data points. As these methods have a different training objective (Target 2, Section 3, Elsemüller et al., 2025) rather than the analytic posterior, we do not include them in other case study comparisons. Instead, we focus on standard simulation-based NPE and state-of-the-art MCMC via Stan, which target the analytic posterior and serve as appropriate baseline and benchmark, respectively, for evaluating our semi-supervised NPE+SC approach.

## E    DETAILED SETUP OF THE AIR TRAFFIC CASE STUDY

The parameters $\alpha_j$ are country-level intercepts, $\beta_j$ are the autoregressive coefficients, $\gamma_j$ are the regression coefficients of household debt and $\delta_j$ are the regression coefficients of GDP per capita, and $\sigma_j$ is the standard deviation of the noise term. This model was previously used within ABI in Habermann et al. (2024). As commonly done for autoregressive models, we regress on time period differences to mitigate non-stationarity. This is critical for simulation-based inference because when $\beta_j > 1$, exponential growth quickly produces unrealistic air traffic volumes. Moreover, amortizing over covariate spaces, such as varying GDP per capita between countries, can lead to model misspecification if such fluctuations are underrepresented in training.

For the air traffic model defined in Section 4.2, we set independent priors on the parameters as follows:

$$
\begin{aligned}
\alpha_j &\sim \text{Normal}(0, 0.5) & \beta_j &\sim \text{Normal}(0, 0.2) \\
\gamma_j &\sim \text{Normal}(0, 0.5) & \delta_j &\sim \text{Normal}(0, 0.5) \\
\log(\sigma_j) &\sim \text{Normal}(-1, 0.5).
\end{aligned}
\tag{20}
$$

For the NPEs $q(\theta \mid x)$, we use a neural spline flow (Durkan et al., 2019) with 6 coupling layers of 128 units each utilizing exponential linear unit activation functions, L2 weight regularization with factor $\gamma = 10^{-3}$, 5% dropout and a multivariate unit Gaussian latent space. These settings were the same

for both the standard simulation-based loss and our proposed semi-supervised loss. The simulation budget was set to $N = 1024$. For the summary network, we use a long short-term memory layer with 64 output dimensions followed by two dense layers with output dimensions of 256 and 64. The inference and summary networks are jointly trained using the Adam optimizer for 100 epochs with a batch size of 32 and a learning rate of $5 \times 10^{-4}$.

# F    COMPREHENSIVE RESULTS FOR THE AIR TRAFFIC CASE STUDY

Table 1: Posterior metrics for NPE and NPE augmented with self-consistency loss (NPE+SC) relative to Stan. For each parameter, absolute bias in posterior means and standard deviations, as well as Wasserstein distance, are reported. Values are shown as mean (SE) across 15 countries. Self-consistency loss was evaluated using $M \in \{4, 8, 15\}$ countries during training.

| Parameter | NPE | NPE+SC ($M = 4$) | NPE+SC ($M = 8$) | NPE+SC ($M = 15$) |
|---|---|---|---|---|
| | | Mean bias ($|\mu - \mu_{\text{Stan}}|$) | | |
| $\alpha$ | $0.079 \pm 0.019$ | $0.003 \pm 0.009$ | $0.014 \pm 0.013$ | $\mathbf{0.002 \pm 0.001}$ |
| $\beta$ | $0.153 \pm 0.024$ | $0.012 \pm 0.026$ | $0.031 \pm 0.023$ | $\mathbf{0.001 \pm 0.002}$ |
| $\gamma$ | $0.087 \pm 0.045$ | $0.048 \pm 0.037$ | $0.006 \pm 0.026$ | $\mathbf{0.002 \pm 0.003}$ |
| $\delta$ | $0.053 \pm 0.033$ | $0.046 \pm 0.033$ | $0.042 \pm 0.023$ | $\mathbf{0.003 \pm 0.004}$ |
| $\log(\sigma)$ | $0.215 \pm 0.076$ | $0.207 \pm 0.076$ | $0.148 \pm 0.058$ | $\mathbf{0.002 \pm 0.002}$ |
| | | Standard deviation bias ($|\sigma - \sigma_{\text{Stan}}|$) | | |
| $\alpha$ | $0.033 \pm 0.011$ | $0.020 \pm 0.008$ | $0.020 \pm 0.007$ | $\mathbf{0.002 \pm 0.001}$ |
| $\beta$ | $0.055 \pm 0.010$ | $0.011 \pm 0.007$ | $0.004 \pm 0.003$ | $\mathbf{0.001 \pm 0.002}$ |
| $\gamma$ | $0.058 \pm 0.015$ | $0.022 \pm 0.016$ | $0.035 \pm 0.013$ | $\mathbf{0.005 \pm 0.003}$ |
| $\delta$ | $0.038 \pm 0.015$ | $0.018 \pm 0.016$ | $0.031 \pm 0.012$ | $\mathbf{0.005 \pm 0.002}$ |
| $\log(\sigma)$ | $0.049 \pm 0.015$ | $0.058 \pm 0.017$ | $0.011 \pm 0.008$ | $\mathbf{0.004 \pm 0.002}$ |
| | | Wasserstein distance | | |
| $\alpha$ | $0.086 \pm 0.019$ | $0.033 \pm 0.009$ | $0.035 \pm 0.013$ | $\mathbf{0.006 \pm 0.001}$ |
| $\beta$ | $0.161 \pm 0.024$ | $0.070 \pm 0.026$ | $0.054 \pm 0.023$ | $\mathbf{0.009 \pm 0.002}$ |
| $\gamma$ | $0.154 \pm 0.045$ | $0.112 \pm 0.037$ | $0.068 \pm 0.026$ | $\mathbf{0.014 \pm 0.003}$ |
| $\delta$ | $0.119 \pm 0.033$ | $0.102 \pm 0.033$ | $0.064 \pm 0.023$ | $\mathbf{0.013 \pm 0.002}$ |
| $\log(\sigma)$ | $0.304 \pm 0.076$ | $0.282 \pm 0.076$ | $0.170 \pm 0.058$ | $\mathbf{0.011 \pm 0.002}$ |

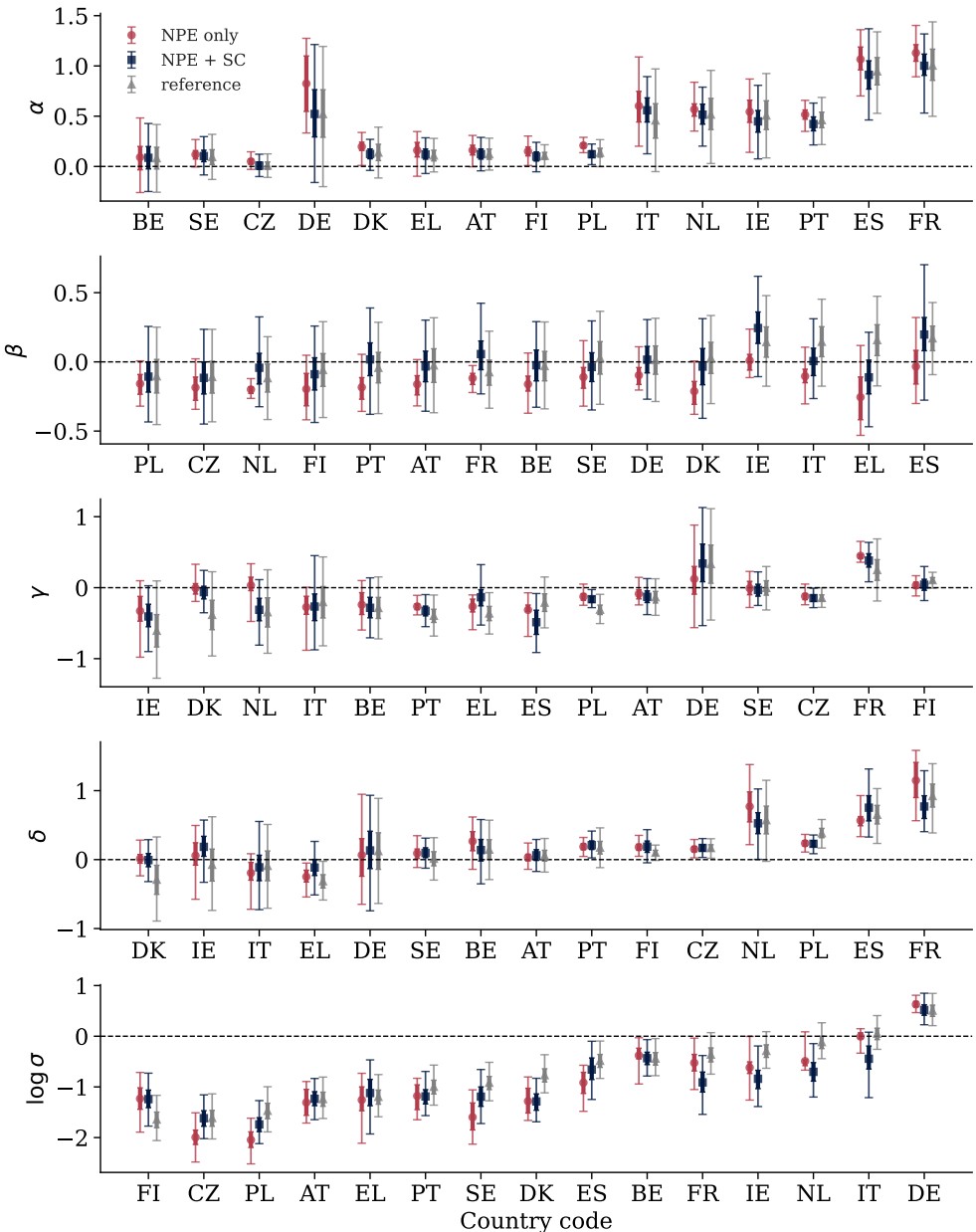

Figure 9: Comparison of posterior estimates between standard amortized NPE (red circles), NPE augmented by our self-consistency loss (NPE + SC; blue squares) and Stan (reference; gray triangles). The plots illustrate central 50% (thick lines) and 95% (thin lines) credible intervals of all five parameters for different countries, sorted by the lower 5% quantile according to Stan. Abbreviations follow the ISO 3166 alpha-2 codes. The self-consistency loss was evaluated on data from $M = 4$ countries during training.

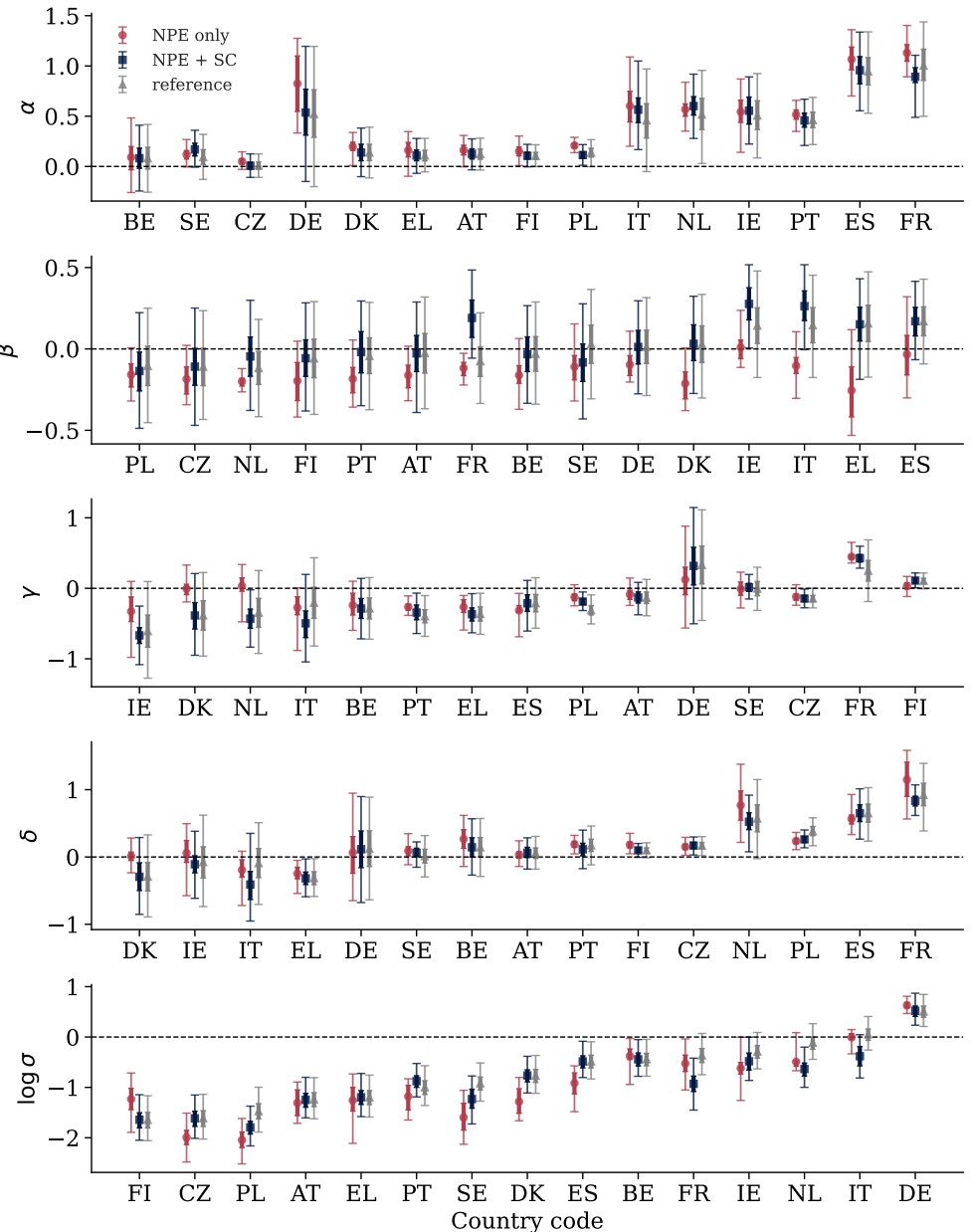

Figure 10: Comparison of posterior estimates between standard amortized NPE (red circles), NPE augmented by our self-consistency loss (NPE + SC; blue squares), and Stan (reference; gray triangles). The plots illustrate central 50% (thick lines) and 95% (thin lines) credible intervals of all five parameters for different countries, sorted by the lower 5% quantile according to Stan. Abbreviations follow the ISO 3166 alpha-2 codes. The self-consistency loss was evaluated on data from $M = \mathbf{8}$ countries during training.

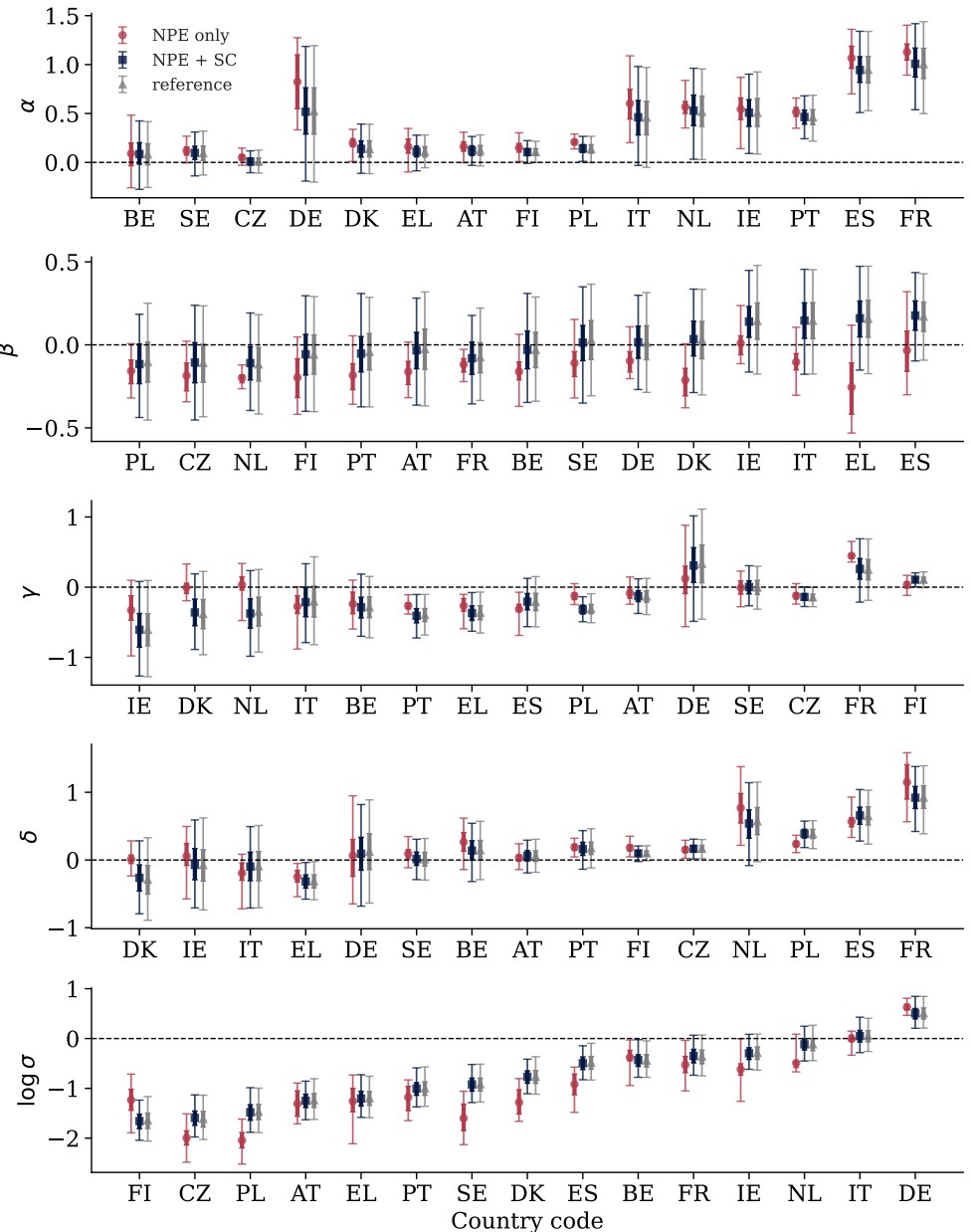

Figure 11: Comparison of posterior estimates between standard amortized NPE (red circles), NPE augmented by our self-consistency loss (NPE + SC; blue squares), and Stan (reference; gray triangles). The plots illustrate central 50% (thick lines) and 95% (thin lines) credible intervals of all five parameters for different countries, sorted by the lower 5% quantile according to Stan. Abbreviations follow the ISO 3166 alpha-2 codes. The self-consistency loss was evaluated on data from $M = \mathbf{15}$ countries during training.

## F.1 COMPARISON TO VAE AND ONLINE TRAINING OF NPE

We conducted additional experiments using a variational auto-encoder (VAE) (Kingma & Welling, 2013) baseline with a known decoder (likelihood) and a neural encoder (posterior network). The encoder is a feed-forward network with three hidden layers of 128 units each, followed by two output heads that parameterize the mean and standard deviation of a standard normal base distribution. The model was trained on 40,960 simulated pairs until convergence. We trained NPE and NPE + SC in online setting that leads to training data of the same size as VAE. We also compare our method against Schmitt et al. (2024) where simulated data is used to evaluate SC loss during training. We similarly used 40,960 simulations for training NPE and 32 simulations for evaluating self-consistency loss for 40 epochs.

Results comparing the above methods for the air-traffic experiment are reported in Table 2 and Figure 12. These results show that the performance of VAE with known likelihood is comparable to NPE (which does not use explicit likelihood density) but it is much worse than our semi-supervised (NPE + SC) approach. Our results also show that the method of Schmitt et al. (2024) does not overcome model misspecification and performs poorly compared to our semi-supervised approach.

Table 2: Posterior Wasserstein distances, mean bias, and standard deviation bias for online trained NPE, Schmitt et al. (2024), VAE, and our proposed NPE + SC method relative to Stan. The self-consistency loss was evaluated on data from $M = \mathbf{15}$ countries during training. Metrics are averaged over all 15 countries.

| Parameter | Mean bias ($|\mu - \mu_{\text{Stan}}|$) | | | |
| --- | --- | --- | --- | --- |
| | NPE (online) | Schmitt et al. (2024) | VAE | NPE (online) + SC |
| $\alpha$ | $0.075 \pm 0.028$ | $0.026 \pm 0.019$ | $0.060 \pm 0.017$ | $\mathbf{0.008 \pm 0.002}$ |
| $\beta$ | $0.087 \pm 0.020$ | $0.052 \pm 0.028$ | $0.096 \pm 0.020$ | $\mathbf{0.013 \pm 0.003}$ |
| $\gamma$ | $0.101 \pm 0.014$ | $0.200 \pm 0.071$ | $0.108 \pm 0.020$ | $\mathbf{0.014 \pm 0.003}$ |
| $\delta$ | $0.096 \pm 0.018$ | $0.112 \pm 0.072$ | $0.110 \pm 0.028$ | $\mathbf{0.009 \pm 0.002}$ |
| $\log(\sigma)$ | $0.189 \pm 0.051$ | $0.042 \pm 0.141$ | $0.142 \pm 0.024$ | $\mathbf{0.009 \pm 0.002}$ |
| | Standard deviation bias ($|\sigma - \sigma_{\text{Stan}}|$) | | | |
| $\alpha$ | $0.030 \pm 0.008$ | $0.003 \pm 0.021$ | $0.029 \pm 0.006$ | $\mathbf{0.005 \pm 0.001}$ |
| $\beta$ | $\mathbf{0.011 \pm 0.002}$ | $0.039 \pm 0.039$ | $0.050 \pm 0.004$ | $\mathbf{0.011 \pm 0.002}$ |
| $\gamma$ | $0.058 \pm 0.016$ | $0.022 \pm 0.033$ | $0.079 \pm 0.014$ | $\mathbf{0.009 \pm 0.002}$ |
| $\delta$ | $0.059 \pm 0.017$ | $0.001 \pm 0.041$ | $0.083 \pm 0.015$ | $\mathbf{0.016 \pm 0.003}$ |
| $\log(\sigma)$ | $0.170 \pm 0.009$ | $0.066 \pm 0.034$ | $0.026 \pm 0.006$ | $\mathbf{0.013 \pm 0.002}$ |
| | Wasserstein distance | | | |
| $\alpha$ | $0.080 \pm 0.028$ | $0.068 \pm 0.017$ | $0.065 \pm 0.016$ | $\mathbf{0.010 \pm 0.002}$ |
| $\beta$ | $0.088 \pm 0.020$ | $0.133 \pm 0.024$ | $0.102 \pm 0.019$ | $\mathbf{0.018 \pm 0.002}$ |
| $\gamma$ | $0.113 \pm 0.015$ | $0.278 \pm 0.050$ | $0.125 \pm 0.019$ | $\mathbf{0.018 \pm 0.003}$ |
| $\delta$ | $0.110 \pm 0.020$ | $0.240 \pm 0.049$ | $0.128 \pm 0.027$ | $\mathbf{0.017 \pm 0.002}$ |
| $\log(\sigma)$ | $0.227 \pm 0.045$ | $0.459 \pm 0.073$ | $0.144 \pm 0.024$ | $\mathbf{0.016 \pm 0.002}$ |

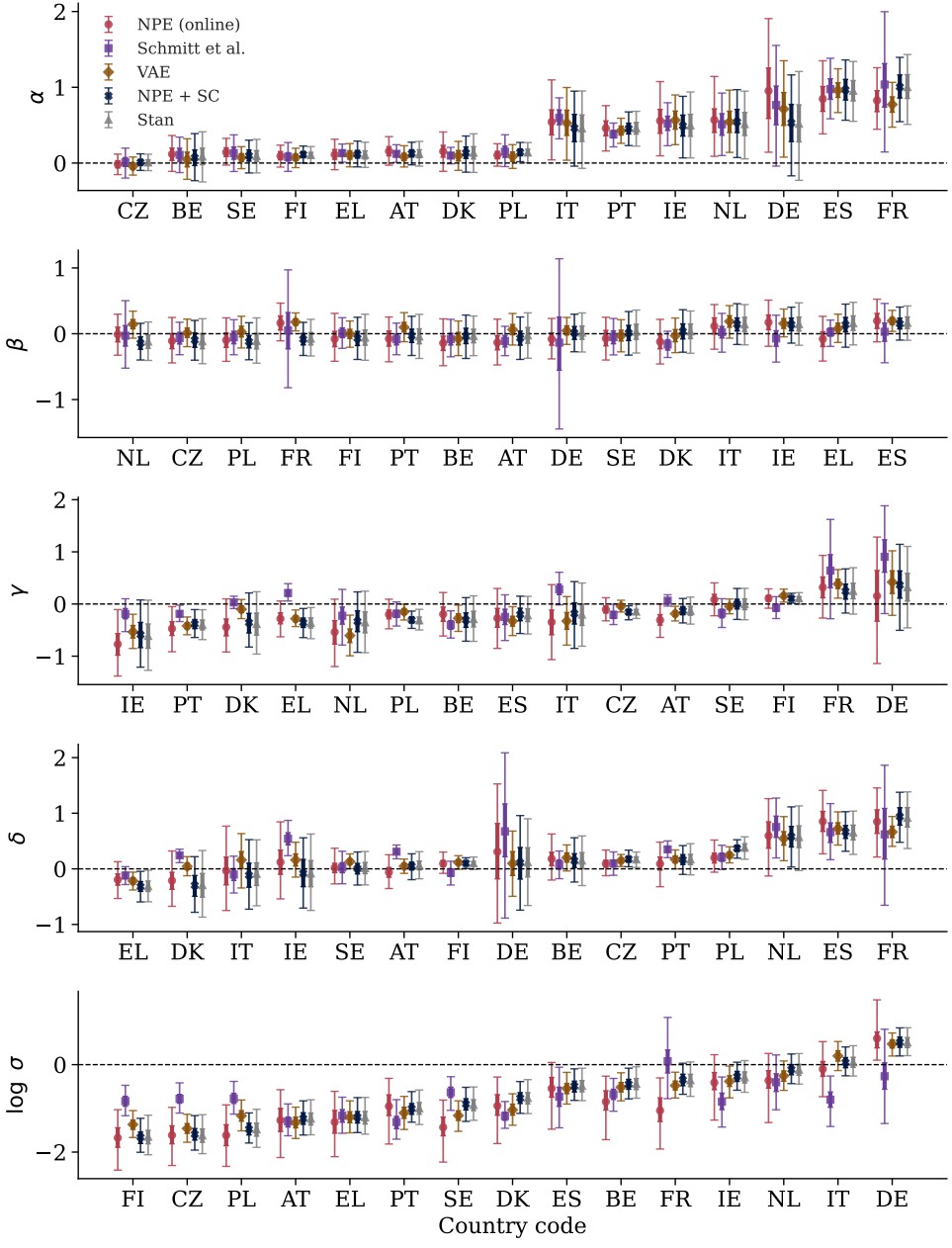

Figure 12: Comparison of posterior estimates among online trained NPE, Schmitt et al. (2024), VAE, and our proposed semi-supervised NPE+SC method against Stan (reference). The self-consistency loss was evaluated on data from $M = \mathbf{15}$ countries during training. The results clearly show that our method is consistent with Stan and outperforms all other methods by a large margin.

## G   Detailed setup of the neuron activation case study

### G.1   Model description

The prior and likelihood are given by

$$g_{\text{Na}} \sim \text{LogNormal}(\log(110), 0.1^2), \; g_{\text{K}} \sim \text{LogNormal}(\log(36), 0.1^2), \; g_{\text{M}} \sim \text{LogNormal}(\log(0.2), 0.5^2)$$

$$E_{\text{Na}} \sim \text{Normal}(50, 5^2), \; E_{\text{K}} \sim \text{Normal}(-77, 5^2), \; E_{\text{leak}} \sim \text{Normal}(-55, 5^2), \; C_m \sim \text{Normal}(1, 0.05^2)$$

$$y_{i,t} \sim \text{Student-t}(V_m(t), 0.1^2, \text{df=10}),$$

where $g_{\text{Na}}, g_{\text{K}}, g_{\text{M}}$ denote the maximum conductances (in mS/cm$^2$) of sodium and two different types of potassium channels; $E_{\text{Na}}, E_{\text{K}}, E_{\text{leak}}$ are the sodium, potassium, and leak reversal potentials (in mV) for sodium, potassium, and leak currents; $C_m$ is the membrane capacitance (in $\mu$F/cm$^2$). The membrane voltage $V_m(t)$ at time $t$ is obtained by solving a set of five ordinary differential equations. Here, $X \sim \text{LogNormal}(\mu, \sigma^2)$ denotes a log-normal distribution with $\log(X) \sim \text{Normal}(\mu, \sigma^2)$.

The membrane voltage $V_m(t)$ evolves according to the classical Hodgkin-Huxley model, extended with an additional slow (muscarinic, M-type) potassium channel with current $I_M$. The total current across the membrane is modeled as:

$$C_m \frac{dV_m}{dt} = I_{\text{Na}} + I_{\text{K}} + I_{\text{M}} + I_{\text{leak}} + I_{\text{in}}(t), \tag{21}$$

where $I_{\text{Na}}, I_{\text{K}}, I_{\text{M}}, I_{\text{leak}}$ are the sodium, potassium, M-type potassium, and leak currents respectively; and $I_{\text{in}}(t)$ is an externally applied current, which is set to be a pulse input of 3.248 nA between $t_{\text{on}} = 10$ms and $t_{\text{off}} = 50$ms. The ionic currents are calculated as:

$$I_{\text{Na}} = g_{\text{Na}} m^3 h (E_{\text{Na}} - V_m) \tag{22}$$

$$I_{\text{K}} = g_{\text{K}} n^4 (E_{\text{K}} - V_m) \tag{23}$$

$$I_{\text{M}} = g_{\text{M}} p (E_{\text{K}} - V_m) \tag{24}$$

$$I_{\text{leak}} = g_{\text{leak}} (E_{\text{leak}} - V_m), \tag{25}$$

where the leak conductance is fixed to $g_{\text{leak}} = 0.1$mS/cm$^2$; and $m$, $h$, $n$, $p$ are gating variables. The gating variables take the form:

$$\frac{dx}{dt} = \frac{x_\infty(V_m) - x}{\tau_x(V_m)}, \qquad x \in \{n, m, h, p\}, \tag{26}$$

with $x_\infty(V_m) = \alpha_x(V_m)/(\alpha_x(V_m) + \beta_x(V_m))$, and $\tau_x(V_m) = 1/(\alpha_x(V_m) + \beta_x(V_m))$ for $n$, $m$ and $h$, where $\alpha_x$ and $\beta_x$ are voltage-dependent rate functions defined as:

$$\alpha_n(V_m) = \frac{0.032 \cdot \exp(-0.2(V_m - 75))}{0.2}, \quad \beta_n(V_m) = \frac{0.28 \cdot \exp(0.2(V_m - 100))}{0.2},$$

$$\alpha_m(V_m) = \frac{0.32 \cdot \exp(-0.25(V_m - 73))}{0.25}, \quad \beta_m(V_m) = \frac{0.28 \cdot \exp(0.2(V_m - 100))}{0.2},$$

$$\alpha_h(V_m) = 0.128 \cdot \exp(\frac{-(V_m - 77)}{18}), \qquad \beta_h(V_m) = \frac{4}{1 + \exp(-0.2(V_m - 100))}$$

For the gating variable $p$ of the M-type potassium channel, a sigmoidal steady-state activation and custom time constant are used:

$$p_\infty(V_m) = \frac{1}{1 + \exp(-0.1(V_m + 35))}, \quad \tau_p(V_m) = \frac{600}{3.3 \cdot \exp(0.05(V_m + 35)) + \exp(-0.05(V_m + 35))}$$

Numerical integration of the system is performed using a fixed-step Euler method over a time window $[0, 60]$ with time step $\Delta t = 0.01$. Voltage traces $V_m(t)$ are downsampled to every 30th observation, and 200-dimensional time series are simulated according to $y_{i,t} \sim \text{Student-t}(V_m(t), 0.1^2, \text{df}=10)$.

### G.2 NETWORK ARCHITECTURE AND TRAINING

For the NPEs $q(\theta|y_{i,t})$, we use a neural spline flow (Durkan et al., 2019) with 10 coupling layers of 256 units each utilizing ReLU activation functions, L2 weight regularization with factor $\gamma = 10^-3$, 5% dropout and a multivariate unit Gaussian latent space. These settings were the same for both the standard simulation-based loss and our proposed semi-supervised loss. For the summary network, we use a long short-term memory layer with 100 output dimensions followed by a sequence of dense layers with output dimensions of 400, 200, 100, and 50, respectively. The inference and summary network are jointly trained using the Adam optimizer with a batch size of 256 for 100 epochs and a fixed learning rate of $5 \times 10^{-4}$, followed by 100 epochs with a fixed learning rate of $5 \times 10^{-5}$ and a final run of 100 epochs with a learning rate of $5 \times 10^{-6}$.

## H COMPREHENSIVE RESULTS OF THE NEURON ACTIVATION CASE STUDY

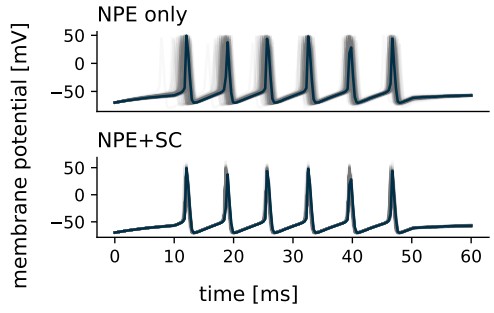

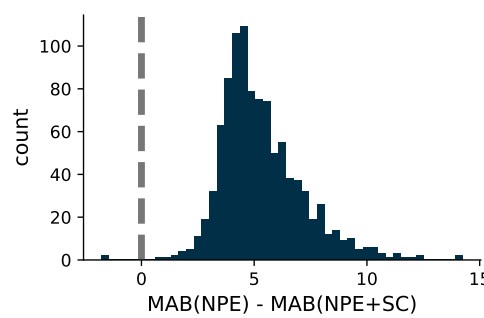

**(a):** Posterior predictive samples without (top row) and with (bottom row) self-consistency loss.

**(b):** Quantitative evaluation of predictive bias.

Figure 13: (a) Posterior predictive samples (gray) inferred from an in-simulation dataset (black) with parameters $\theta \sim \text{Normal}(0, 1)$. Both NPE only and NPE+SC produce predictions that are consistent with the observed data. However, samples from NPE+SC are much closer to the ground truth. (b) Histogram of the mean absolute bias (MAB) difference of posterior predictions computed for 1000 out-of-simulation datasets. NPE+SC has lower bias than NPE for almost all datasets. Mean absolute bias is defined as $\text{MAB}(y_{i,t}, \hat{y}_{i,t}) = \frac{1}{T} \sum_{t=1}^{T} |y_{i,t} - \hat{y}_{i,t}|$ for a time series $y_{i,t}$ with observation index $i$ at time $t = 1, \ldots, T$. $\hat{y}_{i,t} = \frac{1}{S} \sum_{s=1}^{S} p(y_{i,t}|\theta^{(s)})$ denotes the mean of the posterior predictive distribution at time $t$ computed over $S$ posterior samples.

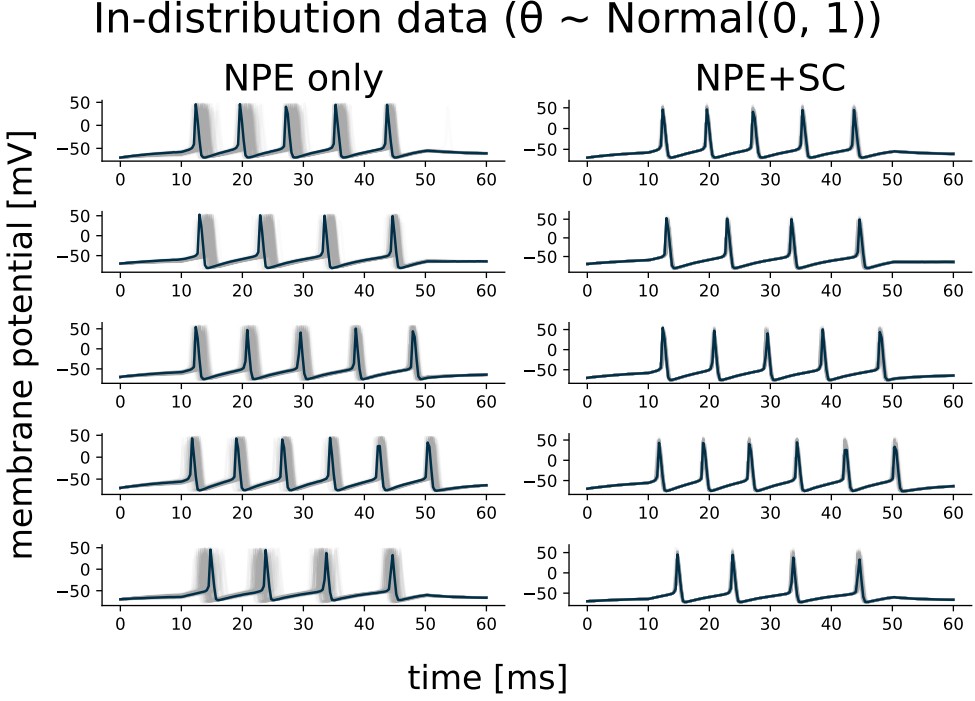

Figure 14: Posterior predictive samples (gray) inferred from 5 simulation datasets following the same distribution as the training data. Both NPE only and NPE+SC show predictions that are consistent with the observed data.

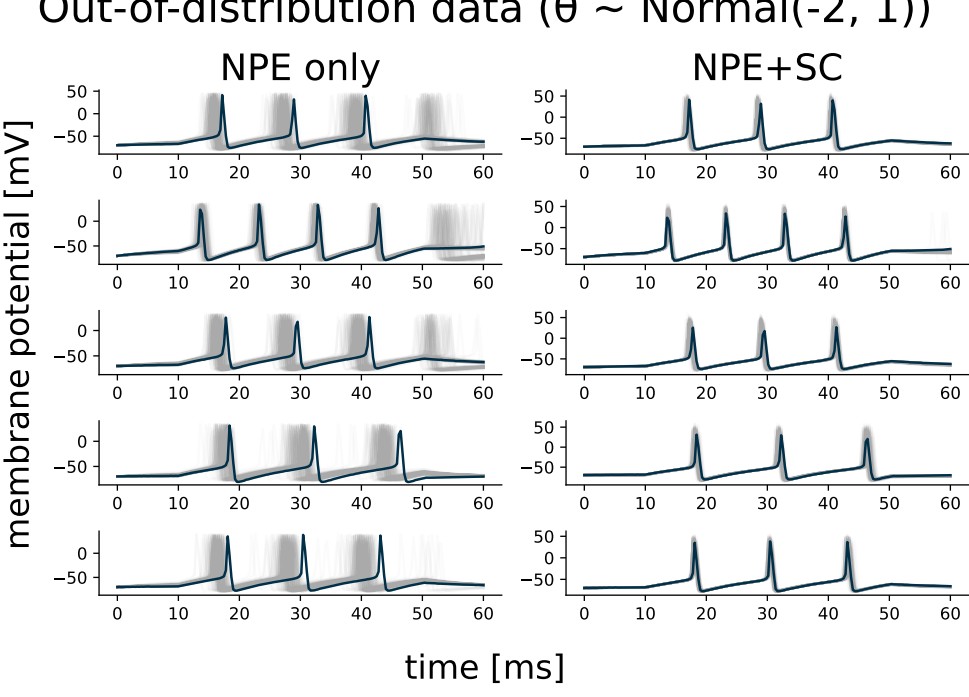

Figure 15: Posterior predictive samples (gray) inferred from 5 out-of-simulation datasets, generated from parameter draws $\theta \sim \text{Normal}(-2, 1)$. While NPE only produces highly biased predictions, NPE+SC is consistent with the observed data.

## I DETAILED SETUP OF THE MNIST IMAGE DENOISING CASE STUDY

We implement the jointly amortized posterior and likelihood networks (Radev et al., 2023) using two normalizing flows with fully connected affine-coupling layers that operate on the flattened 784-pixel vectors. We use the same network architecture that was used by Radev et al. (2023). As both $\theta$ and $x$ are images whose intrinsic dimensionality is significantly lower than their raw pixel count, we use identical 4-layer convolutional neural networks as summary networks for both posterior and likelihood networks. These summary networks terminate in a global average-pooling layer to produce a 128-dimensional summary of the original or blurred image, respectively. The posterior network itself is implemented as a conditional invertible neural network (cINN) consisting of 12 conditional affine-coupling layers; each coupling layer embeds its conditional information via an internal fully connected network with a single hidden layer of 512 units and ReLU activations. The likelihood network adopts exactly the same conditional-coupling architecture. In both cases, we employ a multivariate Student-T distribution in the latent space (Alexanderson & Henter, 2020; Radev et al., 2023), which enables more stable maximum-likelihood training at elevated learning rates.

The prior network that was used to generate blurred MNIST simulations followed the same architecture as the posterior network defined above. We applied the Gaussian blur with PSF = 1.0 to the $5,923$ images of digit "0" in the MNIST training dataset to train the prior network using Adam optimizer for 120 epochs with a batch size of 32, learning rate of $1 \times 10^{-3}$, and a 15% dropout. After training, we generated 12000 blurred images ($\theta$) of the digit 0 to train the posterior and likelihood networks. A Gaussian blur with PSF = 1.0 was further applied to these images to generate observations ($x$) which represent images from a noisy camera. The posterior and likelihood networks along with summary networks were jointly trained on $\{\theta^i, x^i\}_{i=1}^{12000}$ pairs for 100 epochs with a batch size of 32 using a learning rate of $1 \times 10^{-4}$, and a 15% dropout.

For self-consistency loss, the MNIST test set of digit "0" was divided into two subsets comprising 400 and 580 images respectively. The subset with 400 images was used for training self-consistency loss. A Gaussian blur with PSF = 1.0 was applied to these images to generate observations ($x^*$). No prior blur was applied to these images. This represents a prior misspecification scenario as the simulated images used to train NPLE were already blurred before applying the noisy camera while the MNIST images used for inference do not have a prior blur. This misspecification scenario depicts the effectiveness of utilising self-consistency loss to overcome prior misspecification. The self-consistency loss was activated at epoch 21, with its weight linearly ramped from zero to one by epoch 40. Training was performed using minibatches of 16 images, and 32 consistency samples were drawn to estimate the variance. The inference was performed on the other held-out subset comprising 580 images and all the results in Figures 5, 16 and 17 use the MNIST images from this subset.

# J    COMPREHENSIVE RESULTS OF THE MNIST IMAGE DENOISING CASE STUDY

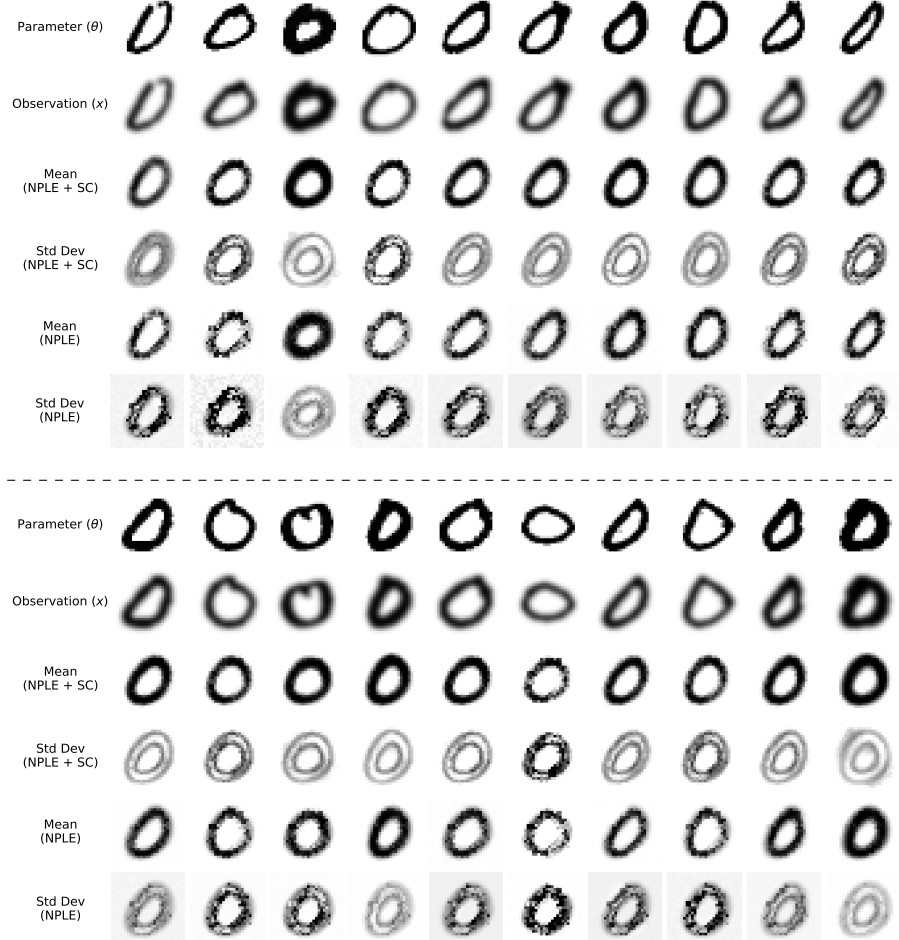

Figure 16: Examples of denoising results for MNIST images of digit "0" in the held-out test set. The *first row* shows ten randomly selected MNIST images ($\theta$), the *second row* depicts the same images after applying the Gaussian blur ($x$), *third* and *fourth rows* depict the mean and standard deviation of 500 posterior samples estimated from the corresponding blurry observations using NPLE + SC based model, and the *fifth* and *sixth rows* depict the mean and standard deviation of 500 posterior samples from model based on NPLE only. Incorporating self-consistency loss significantly improves denoising as the means of reconstructed unblurred image are smoother, less-pixelated and better resemble the ground truth. The darker regions in the standard deviation show the regions of higher variability in the outputs. The standard deviation maps of NPLE + SC based approach are far more coherent showing high variability only along the edges.

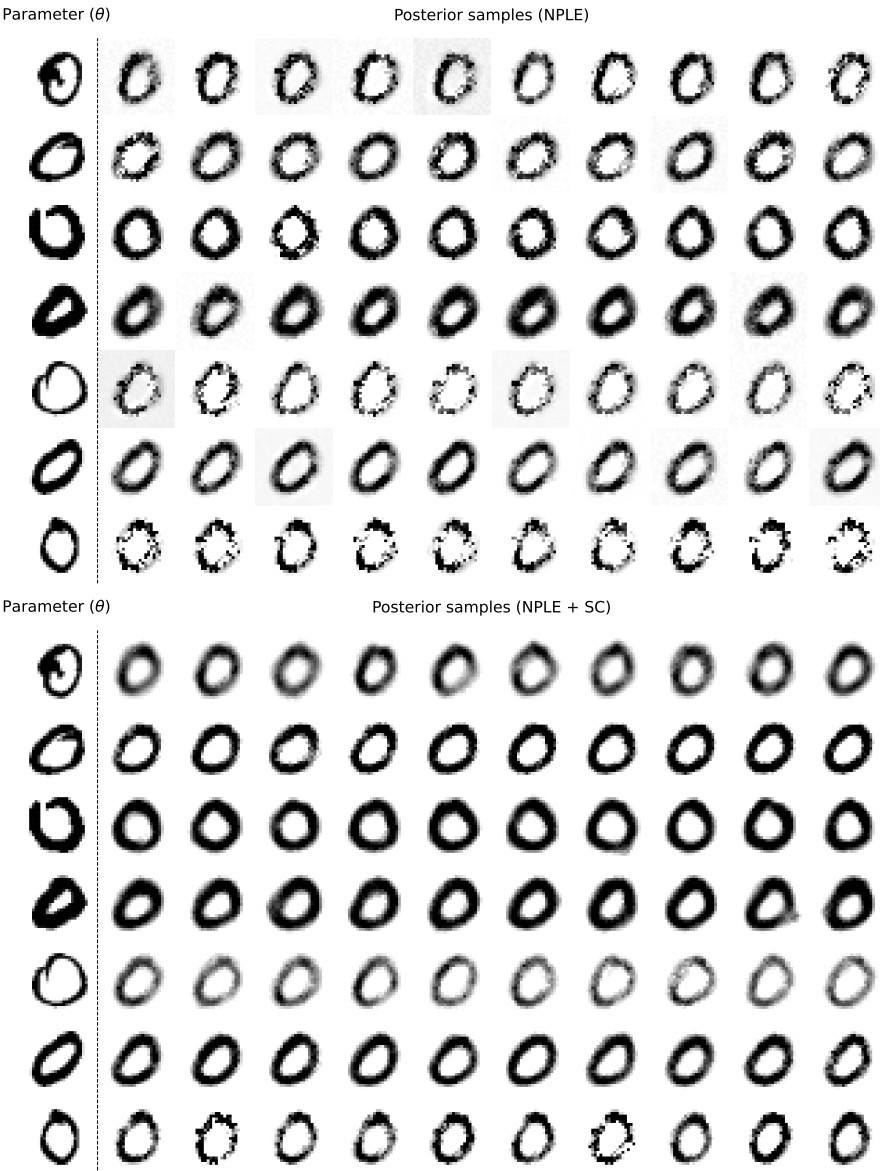

Figure 17: Ground-truth images and the corresponding posterior draws for seven randomly selected MNIST "0" digits from the held-out MNIST test set. In each row, the leftmost panel shows the true image ($\theta$), and the following panels show ten independent samples from the approximate posterior. The top-figure shows the posterior draws using the standard NPLE based model and the bottom figure shows posterior draws from combining self-consistency loss to the NPLE based model. It can clearly be seen that NPLE+SC posterior draws are a better reconstruction of the original image whereas NPLE based posterior samples are highly pixelated.

## K    COMPUTATIONAL RESOURCES FOR EXPERIMENTS

1. **Multivariate normal model:** The experiments were run on a 16-core AMD Ryzen 5950x CPU, equipped with 32 GB of system RAM.

2. **Air traffic case study:** The experiments were conducted on a single MacBook Pro (M3, 2024) equipped with Apple's M3 chip and 16 GB of unified RAM, running macOS Sonoma 14.6. We did not use the GPU cores.

3. **Neuron activation case study:** The experiments were run on a 16-core AMD Ryzen 5950x CPU, equipped with 32 GB of system RAM.

4. **MNIST image denoising:** The experiments were run on a high-performance compute cluster using a GPU-equipped compute node featuring a single NVIDIA Tesla P100-PCIE with 12 GB of dedicated HBM2 memory, paired with 16 GB of system RAM. The training for the longest experiment took $\sim 100$ minutes.

