# OpenReview forum: "Robust Amortized Bayesian Inference with Self-Consistency Losses on Unlabeled Data"
_ICLR.cc/2026/Conference — ICLR 2026 Poster_

### Official Review · Reviewer_ipsv · 2025-10-19

**Soundness:** 3
**Presentation:** 3
**Contribution:** 2
**Rating:** 4
**Confidence:** 3

**Summary:**

The authors propose a new loss term that uses unlabeled data to improve the robustness of amortized neural posterior estimation methods. They thoroughly motivate and explain their loss term and prove that it is a proper loss (in the relevant context of SBI) and present promising results for their method on a simple Gaussian mixture scenario, as well as three more challenging setups (Air traffic data, Hodgkin-Huxley model, MNIST denoising).

**Strengths:**

The core idea of the paper is neat. I also appreciate that it's a simple idea!

Furthermore, the approached problem of robustness is highly relevant in SBI (and beyond). It also seems like the availability of unlabelled real-world data, that is necessary for this approach, is a very reasonable assumption that is often fulfilled in practice. I see this as a major strength of this approach compared to other recent methods.

The overall presentation is good and the structure of the paper very clear and well-written.

Related work is comprehensively discussed.

The idea is original, but some parts of the theory are already present in other papers, albeit for a different purpose.

**Weaknesses:**

Contribution:

I have one major concern with the paper: You do assume for the majority of your theory and experiments that the likelihood under the simulator is known. Doesn't this defeat the purpose of SBI to some degree? If the likelihood is know, why not use a method that directly leverages this information?

I am aware that the case of unknown likelihood + unknown posterior is briefly discussed, but more theoretical insight would be very helpful.

Analogously for the experiments. I believe that the case of unknown likelihood + unknown posterior is the actually relevant one, but it is not really evaluated or thoroughly discussed.

I am also unsure about the benefit of the method if used in combination with post-hoc methods such as SNIS; I believe that many people practically interested in the method might want to use post-hoc methods in combination with this approach.

Presentation:

Please note that those two points are not crucial:

Some parts of the methodology seem a bit repetitive and do not really present very interesting insights. For instance Proposition 4 seems a bit too redundant for this much space. I am wondering if it would help the paper to shorten this section and, for instance, include a brief introduction to amortized SBI in the beginning to help readers less familiar with this field.

Similarly for the description of the multivariate Normal case-study: I feel like it takes up a bit too much space for the practical relevance this case-study has. Instead more results/figures for the other case-studies could be presented in the main body.

Nit: There is a typo around line 392: "enablung"

Experiments:

Please also report metrics for the denoising MNIST case-study.

Seeing C2ST scores for all experiments would also be nice.

While I believe that the experiments overall are similar in scope to what is usually done in SBI, but very small compared to other fields of ML. I would be interested in seeing larger-scale experiments (higher dimensionalities, larger neural networks, more extensive training); but this is not a critical point.

**Questions:**

I would be happy to see your response regarding my concern about the contribution of the paper.

Why should I use SBI, and this method in particular, when I have access to the likelihood under the simulator? Why not directly leverage the likelihood and rely on samples instead?

Are there any further (theoretical) insights into the case where both the likelihood and posterior are unknown?

I am looking forward to your response!

I would be very happy to increase my score if you further justify the applicability of using NPE plus your robustness term when the likelihood is known (i.e. in what case would this be a relevant setup?) OR provide more evidence that using your method with an estimated likelihood is theoretically sound and works well experimentally.

---

> ### Author Response · Authors · 2025-11-20
>
> Dear Reviewer,
>
> We sincerely thank you for taking the time to evaluate our submission and for your thoughtful comments and suggestions. Please find our response below.
>
> **(Contribution) Weakness 1 and 2:**
>
> We are primarily interested in achieving amortization, that is, enabling Bayesian inference in real time after training. The field of Amortized Bayesian Inference (ABI) originates from SBI but has since moved beyond SBI to likelihood-based settings too. While much of our theory and some of our experiments assume access to the simulator likelihood, this does not negate the practical value of ABI with the self-consistency loss. ABI provides substantial amortisation advantages compared to state-of-the-art likelihood-based methods (e.g., MCMC, NUTS) that require costly sampling per observation. SC further improves these benefits by providing accurate posterior estimates under model misspecification.
>
> Even when the likelihood is unknown, we show in multivariate normal and image denoising case studies that SC loss still provides substantial improvements over NPE. It should also be noted that for image denoising experiments, prior, likelihood, and posterior densities are all neural estimated and yet SC shows good performance. However, we would also like to mention that when both posterior and likelihood densities are unknown, self-consistency loss no longer remains strictly-proper. The exact implications of this are likely case dependent and care has to be taken to jointly train the likelihood and posterior networks appropriately.
>
> **(Contribution) Weakness 3:**
>
> For the case studies 1 (multivariate normal) and 4 (image denoising) we have considered the cases where the likelihood is unknown. In the multivariate normal example, we show that even when likelihood is approximated, SC does give robustness benefits. In the image denoising example, we further alleviate the problem and have an approximator for prior as well along with likelihood and posterior networks. Even in such a setting, we can clearly observe the robustness gains from using self-consistency.
>
> **(Contribution) Weakness 4:**
>
> One can indeed easily combine our SC-NPE with post-hoc methods such as importance sampling. Often, importance sampling applied to NPE alone fails because the NPE posterior is too far away from the analytic target posterior (Li et al. 2025). In contrast, the combination of SC-NPE and importance sampling should further increase the range of scenarios in which the analytic posterior of the model can accurately be estimated. In our experiments, SC-NPE alone was already sufficient by itself but in other cases, adding post-hoc methods on top could be beneficial. We will add these points in the discussion in the camera-ready version.
>
> **(Presentation) Weakness 5:**
>
> We appreciate the reviewer’s idea of making the paper more accessible. While we would try not to omit Proposition 4, as it complements our theoretical exposition, we will definitely add a paragraph on ABI in introduction as we will have an additional page for the camera-ready version.
>
> **(Presentation) Weakness 6:**
>
> We appreciate the reviewer’s concern on presentation of the paper and we will utilise much of the additional page in camera-ready version to depict additional results. If needed, we will cut back the description of the multivariate normal case study to gain even more space.
>
> **(Nit) Weakness 7:** We thank the reviewer for pointing it out. We will correct the typo in the camera-ready version.
>
> **(Experiments) Weakness 8:**  We will report the metrics for MNIST case-study in the camera-ready version.
>
> **(Experiments) Weakness 9:**
>
> We quantitatively compare our amortized posteriors against ground-truth references using the analytic posterior where available and Stan-based MCMC estimates otherwise. We report mean bias, standard deviation bias, and maximum mean discrepancy (MMD) as primary metrics, and include Wasserstein distances in selected cases. Together, these metrics capture both calibration and distributional similarity aspects of posterior accuracy. Since C2ST is effectively another measure of sample distinguishability, adding it would not provide additional insight. We therefore opted for a more interpretable and complementary set of metrics that are standard in ABI.
>
> **(Experiments) Weakness 10:**
>
> In current SBI, multiple hundred parameters (as in our 784D MNIST case study) is already considered very high dimensional, due to the complexities arising in solving probabilistic inverse problems. In fact, most SBI problems tend to have between 2 and 20 parameters only and already these provide major challenges. Of course, a few hundred parameters is not high dimensional at all in the general field of ML, but there also the problems are different. Due to the time constraints of the rebuttal period, we could not construct much higher dimensional case studies, but we fully agree that SBI should gradually move towards larger scale experiments.

---

> > ### Author Response · Authors · 2025-11-20
> >
> > **Question 1:**
> >
> > Even when the likelihood is tractible, ABI and our method in particular, offers practical advantages: Once trained, the neural posterior can perform inference for new observations instantly, without re-running MCMC, which is crucial in scenarios requiring repeated or online inference. ABI methods parallelize naturally across simulators and data, whereas likelihood-based MCMC often becomes computationally infeasible when many re-runs are required. As a result, through ABI, we gain tremendous speed improvements at inference time compared to state-of-the-start sampling methods such as MCMC. This aspect is also showcased in other papers, for example, in Li et al. 2025.
> >
> > **Question 2:**
> >
> > As we discuss in our paper, when both posterior and likelihood densities are unknown, the self-consistency loss is no longer strictly-proper. The exact implications of this are likely case dependent and care has to be taken to jointly train the likelihood and posterior networks appropriately. Empirically, we saw good performance in this setting in our multivariate normal and image denoising case studies, making us hopeful that we can achieve improved performance in the estimated likelihood and posterior case also more generally.
> >
> > We hope that our responses adequately address the concerns and clarify any misunderstandings. We are happy to engage in further discussion if needed, and we respectfully hope that you will consider updating your evaluation in light of our clarifications.
> >
> > Best wishes,
> >
> > The authors
> >
> > **References:**
> > 1. Li, Chengkun, et al. "Amortized bayesian workflow." arXiv preprint arXiv:2409.04332 (2024).

---

> > ### Comment · Reviewer_ipsv · 2025-11-21
> >
> > Dear authors,
> >
> > Thank you for your thoughtful and detailed response.
> >
> > I totally agree with your point that traditional MCMC methods are not capable of amortization, which can be a drawback; and the whole point of ABI is to do precisely that amortization.
> >
> > However, there are various methods that can do amortization while taking into account a possibly known likelihoods, one of them being a VAE.
> >
> > Using a VAE you would have the following objective, right?
> >
> > $$L = E_{p(x)} [ - KL(q(\theta|x)||p(\theta)) + E_{q(\theta|x)}[\log p(\theta|x)]]$$
> >
> > So a VAE can accomplish, given the log likelihood (and log-prior), exactly what you want while directly using the likelihood **and** amortizing, right?
> > I believe that using a VAE, especially a "modern", improved version of the original approach, would be a very attractive alternative to your method, in the know-likelihood-case.
> >
> > However, I also feel like your "self-consistency" objective is not actually so different from the ELBO objective. Could you elaborate?

---

> ### Author Response · Authors · 2025-11-22
>
> Dear Reviewer,
>
> Thank you for your question.
>
> You are right that the variance-based SC loss (Prop. 2) is related to reverse KL divergence, as we can decompose
> $$
> Var_q[log(q(θ)/p(θ))] = E_q[(log(q(θ)/p(θ)))^2] - (E_q[log(q(θ)/p(θ))])^2,
> $$
> where the second term is the square of the reverse KL divergence. This is also described in Köthe (2023) and motivated in Schmitt et al., (2024). Prompted by your good observation, we will make sure to highlight this connection more thoroughly. For our current paper, the importance is that we combine simulation-based losses with likelihood-based losses. If we just used the likelihood-based losses alone, we would need much more real-world data than a handful to generalize over relevant spaces. So the fact that amortization with neural networks works well in the first place is because we can easily generate thousands of simulated datasets on which to train a (supervised) simulation-based loss to then refine with (unsupervised) likelihood-based loss functions for the few (unlabeled) real-world datasets we have available. The combination of different VAE losses and SBI losses is definitely very interesting and we will explore this in future research. Due to time and space restrictions, these method developments and corresponding experiments cannot fit into our current paper, which already features four fully-fledged experiments in addition to the ones in the Appendix.
>
> Best wishes,
>
> The authors
>
> References:
> 1. Köthe, Ullrich. "A review of change of variable formulas for generative modeling." arXiv preprint arXiv:2308.02652 (2023).
> 2. Schmitt, Marvin, et al. "Leveraging Self-Consistency for Data-Efficient Amortized Bayesian Inference." Forty-first International Conference on Machine Learning.

---

> > ### Comment · Reviewer_ipsv · 2025-11-22
> >
> > Thank you for your explanation!
> >
> > I agree with your argument that you would not want to train with the likelihood-based loss exclusively on large amounts of real-world data, since it might not be available.
> >
> > But would the authors agree or disagree that, for the case of available likelihood, using it in a VAE that is trained on the simulated data would be a better way to approach the given problem? (The consistency-loss could be used on top of this).
> > If not, why?
> >
> > Please also note that I am aware that setting up experiments to test a VAE-parametrization, using the known likelihood, of the posterior would go beyond the scope of the rebuttal at this.
> >
> > In case you agree that the VAE-parametrization would be a nice alternative in the known-likelihood-case, please highlight it for the camera-ready version, since it might be very helpful for people working in setups with known likelihood.

---

> > > ### Author Response · Authors · 2025-11-23
> > >
> > > Dear Reviewer,
> > >
> > > We agree that a VAE-parameterization is an interesting alternative and we would highlight potential avenues in the camera-ready version, with a specific eye on variational methods for SBI. In a way, normalizing flows can be viewed as a VAE with a flexible posterior and a bijective mapping between latents and targets (Köthe, 2023), so we agree that making these connections given the additional space will be helpful for generating new ideas and thank you for initiating this stimulating discussion!
> > >
> > > Best wishes,
> > >
> > > The authors

---

> > > > ### Author Response · Authors · 2025-11-27
> > > >
> > > > Dear Reviewer,
> > > >
> > > > We conducted additional experiments using a VAE baseline with a known decoder (likelihood) and a neural encoder (posterior network). The encoder is a feed-forward network with three hidden layers of 128 units each, followed by two output heads that parameterize the mean and standard deviation of a standard Normal base distribution. The model was trained on 40,960 simulated pairs until convergence. We trained NPE and NPE + SC again in online setting that leads to training data of the same size as VAE. Results for the air-traffic experiment (Case Study 2) are reported in the table below:
> > > >
> > > > WD: Wasserstein distance w.r.t Stan
> > > >
> > > > | Parameter      | WD  (VAE)            | WD (NPE)            |  WD (NPE+SC)            |
> > > > |----------------|------------------------------|------------------------------|-------------------------------|
> > > > | $\alpha$     | $0.065 \pm 0.016$          | $0.080 \pm 0.028$          | $0.010 \pm 0.002$           |
> > > > | $\beta$      | $0.102 \pm 0.019$          | $0.088 \pm 0.020$          | $0.018 \pm 0.002$           |
> > > > | $\gamma$     | $0.125 \pm 0.019$          | $0.113 \pm 0.015$          | $0.018 \pm 0.003$           |
> > > > | $\delta$     | $0.128 \pm 0.027$          | $0.110 \pm 0.020$          | $0.017 \pm 0.002$           |
> > > > | $\log \sigma$| $0.144 \pm 0.024$          | $0.227 \pm 0.045$          | $0.016 \pm 0.002$           |
> > > >
> > > > These results show that a VAE with a known likelihood performs similarly to NPE (which does not rely on an explicit likelihood), but both are substantially outperformed by our semi-supervised method (NPE + SC). We will include these comparisons in the camera-ready version. Importantly, the results also indicate that combining the SC loss with a simulation-trained VAE that has access to an explicit likelihood offers no clear advantage: NPE already achieves comparable performance, is computationally more efficient, and scales better to higher-dimensional settings. Moreover, normalizing flow based posteriors provide far more expressive densities than standard VAEs, which further limits the usefulness of VAE-based approaches in complex inference problems.
> > > >
> > > > Best wishes,
> > > >
> > > > The authors

---

> > > > > ### Comment · Reviewer_ipsv · 2025-11-28
> > > > >
> > > > > Thank you for conducting those additional experiments; I find them very interesting.
> > > > >
> > > > > I will reflect your efforts in my final evaluation of the paper.

---

### Official Review · Reviewer_L5ck · 2025-10-31

**Soundness:** 3
**Presentation:** 3
**Contribution:** 3
**Rating:** 6
**Confidence:** 3

**Summary:**

This paper proposes a semi-supervised approach to amortized Bayesian inference using self-consistency losses on unlabeled data, demonstrating improved robustness in out-of-distribution settings. While the theoretical contributions are solid, key claims about high-dimensionality and robustness are not adequately supported by the experimental evidence.

**Strengths:**

Improving robustness of SBI to data distribution shifts in an important problem.

**Weaknesses:**

- The abstract uses "bad pre-asymptotic behavior" without definition, making the core problem inaccessible.

- When would practitioners have unlabeled real data but not be able to generate more simulations?

- The paper claims "high-dimensional" capability but experiments max out at 100 parameters with significant performance degradation (Figure 2a shows MMD increasing substantially). The MNIST example (784D) is modest by modern standards. Please scale back claims.

- Head-to-head comparisons on all case studies are essential. Also missing: ablations on the choice of p_C(theta), sensitivity to lambda schedule, and performance with different amounts of unlabeled data.

- The paper claims to maintain ABI's speed but provides zero timing comparisons, memory usage, or analysis of overhead from computing SC loss and sampling q_t(theta|x) during training.

- Propositions 1-4 establish strict properness under idealized conditions (universal approximators, infinite data), but there's no finite-sample analysis or characterization of approximation error with realistic network capacities. The gap between asymptotic guarantees and finite-sample behavior needs some comments.

- The MNIST prior misspecification (double-blur vs single-blur) is artificial. How does the method perform under realistic misspecification where the error structure is unknown during training?

**Questions:**

See above.

---

> ### Author Response · Authors · 2025-11-20
>
> Dear Reviewer,
>
> We sincerely thank you for taking the time to evaluate our submission and for your thoughtful comments and suggestions. Please find our response below.
>
> **Weakness 1:**
>
> We thank the reviewer for pointing this out. The term “bad pre-asymptotic behavior” refers to the tendency of current neural posterior estimators to produce biased or overconfident posteriors when trained with limited simulations, before reaching the asymptotic regime of infinite data or model capacity. We have clarified this in the revised abstract.
>
> “This is mainly due to the poor pre-asymptotic behavior of current neural posterior estimators, i.e., their tendency to produce unreliable posteriors when trained with a finite number of simulations and limited model capacity."
>
> **Weakness 2:**
>
> This would be in the abundant cases where the simulator is very expensive (Delaunoy et al. 2024, Lingyi et al. 2025) or the training set contains a fixed number of pre-simulated cases (Elsemüller et al. 2024, Diffenbaugh & Barnes 2023, Orozco et al. 2024), In many of these settings (e.g., Lingyi et al.), there are many more unlabeled real observations than labeled simulations, and this is a common theme for many inverse problems considered difficult to simulate.
>
> In the context of statistical inference, amortized or not, almost every real data set is unlabeled, since we do not know their ground truth parameters; hence the need to perform inference in the first place. And, as shown in most papers about model misspecification in SBI or ABI, more simulations only help with inference accuracy in the typical set of the simulated data. Any inference performed on data beyond this typical set (i.e., many real-world datasets) is likely to be biased or otherwise inaccurate simply because the neural network was not trained in that space.
>
> **Weakness 3:**
>
> You are right that, in the larger field of machine learning, 784D is not high-dimensional at all. However, within the field of SBI and ABI, 784D is in fact very high dimensional (e.g., Dellaporta et al. 2022, Gloeckler et al. 2023, Ward et al. 2022, Wehenkel et al. 2024, Elsemüller et al. 2025). So it depends on the point of comparison. We will better explain and scale down our claims in the camera-ready version.
>
> With regard to the results presented in Figure 2a, we would like to clarify that the data used to minimize the SC loss (centered at $\mu = 3$) differs from the observed data locations used in the evaluation ( $\mu \in [0,10]$). Even in the 100-dimensional setting, the SC estimates remain robust whenever the observed data are close to the region on which the SC loss was trained (i.e., near  $\mu = 3$). Moreover, when the SC loss is trained directly on the observed data distribution, this robustness persists even for  $\mu = 10$. We will include an additional figure in the appendix to illustrate this effect in the camera-ready version.
>
> **Weakness 4:**
>
> *Head-to-head comparisons:* We did in fact already compare our new method to several competing methods. You can find the corresponding results in Appendix D.1.
>  On a conceptual level, none of the existing robustness methods actually target the analytic posterior but some implicitly robustified version. As a result, most of the existing methods perform poorly under prior misspecification and work well only in specific scenarios of likelihood misspecification. What is more, many of these methods assume availability of real labelled data to target the data implied posterior and to the best of our knowledge none of these methods work under minimal available real data.
>
> *Ablation on $p_C(\theta)$:* We conducted preliminary experiments using the prior as the proposal distribution, but this led to instability in the self-consistency loss. Separately, when the neural posterior estimate is used as the proposal during the early stages of NPE training, i.e., while that estimate is still poor or effectively indistinguishable from the prior, the SC loss can blow up, especially in high-dimensional settings. We therefore use scheduling to stabilize training. Once the estimated posterior becomes sufficiently accurate, the SC loss behaves stably and performance improves.We will add some results with the prior as proposal distribution in the appendix to substantiate these claims
>
> *Sensitivity to lambda:* We request the reviewer to refer to Appendix B (lines 711-718) where we have already discussed this question. We will incorporate it into the main text using the additional page for the camera-ready version.
>
> *Different amounts of unlabeled data:* We request the reviewer to refer to Figure 5 (second row) in Appendix D where we have already shown the results for varying amounts of unlabeled data for training the SC loss.
>
>
> *We answer the remaining questions in the following comment due to character limits.*

---

> > ### Author Response · Authors · 2025-11-20
> >
> > **Weakness 5:**
> >
> > In the paper, we specifically talked about the *inference speed* to remain the same. This holds because inference is just a simple (very fast) forward pass through the network. It remains the same regardless of whether or not an SC loss was included during training. That said, the *training time* may increase; in our experiments by a factor of roughly 2 to 2.5. We believe that this is usually a small price to pay for such increased real-world robustness. Memory use was not affected to a relevant degree. We will incorporate the timing comparisons and memory usage in the appendix in the camera-ready version.
> >
> > **Weakness 6:**
> >
> > Thank you for raising this point. Indeed, our theoretical guarantees (Propositions 1–4) hold asymptotically and we will highlight this more clearly in the revised version of the paper. However, we provide empirical evidence of good finite-sample behavior by comparing our learned posteriors with ground-truth estimates obtained via Stan. Across all case studies, the amortized posteriors closely match Stan’s MCMC posteriors, indicating that the proposed method yields accurate inference even under limited data and network capacity. Our case studies also include different training dataset sizes, data modalities, and settings with approximate likelihood. This empirical agreement suggests that finite-sample approximation errors remain small in practice, consistent with our robustness findings.
> >
> > **Weakness 7:**
> >
> > We agree that the MNIST prior misspecification is artificial.  The motive of this case study was to show the applicability of our proposed method to image data where the probability density for prior, likelihood, and posterior is entirely neural approximated. The air traffic experiment (Case study 2) and Hodgin-Huxley experiment (Case study 3) depict the applicability of our method to more realistic misspecification scenarios.
> >
> > We hope that our responses adequately address the concerns and clarify any misunderstandings. We are happy to engage in further discussion if needed, and we respectfully hope that you will consider updating your evaluation in light of our clarifications.
> >
> > Best wishes,
> >
> > The authors
> >
> > **References:**
> > 1. Zhou, Lingyi, et al. "Bridging simulations and observations: New insights into galaxy formation simulations via out-of-distribution detection and Bayesian model comparison-Evaluating galaxy formation simulations under limited computing budgets and sparse dataset sizes." Astronomy & Astrophysics 701 (2025): A44.
> > 2. Delaunoy, Arnaud, et al. "Low-Budget Simulation-Based Inference with Bayesian Neural Networks." arXiv preprint arXiv:2408.15136 (2024).
> > 3. Elsemüller, Lasse, et al. "Sensitivity-aware amortized bayesian inference." arXiv preprint arXiv:2310.11122 (2023).
> > 4. Diffenbaugh, Noah S., and Elizabeth A. Barnes. "Data-driven predictions of the time remaining until critical global warming thresholds are reached." Proceedings of the National Academy of Sciences 120.6 (2023): e2207183120.
> > 5. Orozco, Rafael, et al. "ASPIRE: iterative amortized posterior inference for Bayesian inverse problems." Inverse Problems 41.4 (2025): 045001.
> > 6. Dellaporta, Charita, et al. "Robust Bayesian inference for simulator-based models via the MMD posterior bootstrap." International Conference on Artificial Intelligence and Statistics. PMLR, 2022.
> > 7. Gloeckler, Manuel, et al. "All-in-one simulation-based inference." arXiv preprint arXiv:2404.09636 (2024).
> > 8. Elsemüller, Lasse, et al. "Does Unsupervised Domain Adaptation Improve the Robustness of Amortized Bayesian Inference? A Systematic Evaluation." arXiv preprint arXiv:2502.04949 (2025).
> > 9. Ward, Daniel, et al. "Robust neural posterior estimation and statistical model criticism." Advances in Neural Information Processing Systems 35 (2022): 33845-33859.

---

> > > ### Author Response · Authors · 2025-11-27
> > >
> > > Dear Reviewer,
> > >
> > > We hope that our responses have addressed your concerns. We request you to let us know in case you have additional questions. If you are satisfied with our clarifications, **we would greatly appreciate an updated assessment of our submission**, as we believe our contribution offers substantial value to the ABI/SBI community.
> > >
> > > Best wishes,
> > >
> > > The Authors

---

### Official Review · Reviewer_Lkfz · 2025-10-31

**Soundness:** 2
**Presentation:** 3
**Contribution:** 2
**Rating:** 2
**Confidence:** 4

**Summary:**

The authors propose to include a self-consistency loss regularizer to the typical neural posterior estimation (NPE) class of objectives for amortized inference (more generally, a loss with respect to any score). The regularizer, unlike the main objective, is computed over real data, not simulated draws from the model. The regularizer enforces self-consistency (Eq. 1) on the observed dataset, draws from which are assumed to be the target for inference. The motivation provided by the authors is that this results in more accurate inference, especially in distribution-shift cases where the real observations are substantively different from the simulated data with which $q$ is fit.

**Strengths:**

- The writing and proposed methodology are clear and easy to understand
- The proposed regularizer is intuitive: as the true posterior demonstrates  self-consistency property, it’s a natural extension for the variational posterior to satisfy this condition (approximately)
- The problem setting is a significant one, as distribution shift in simulation-based inference (SBI) problems continues to be a robust area of research.
- The others take care to formalize their result more carefully with respect to strictly proper losses, proving that the regularizer does not alter the problem's solution.

**Weaknesses:**

- The method is combinatorial: the NPE (score-based) objective for SBI is well-established, and Schmitt et al. (2024) introduced the self-consistency loss.
- I don’t find the degree of novelty to be adequate enough to differentiate the work from Schmitt et al. In the case where the simulation model is correct, the proposed method is exactly identical to Schmitt et al. Thus, the authors’ main contribution, to me, seems to be applying the method to the setting of a misspecified simulator. While this is undoubtedly an important setting as acknowledged above, the novelty of applying the method to a new setting is fairly limited
- Experimental evaluation is insufficient. To me, comparing only to NPE isn’t a fair comparison. The experiments involve deliberately mispecifying the model, so no one would expect NPE to perform well. At a minimum, the authors should compare to a VAE trained with the incorrect prior/likelihood (but with the real dataset), as well as the method of Schmitt et al., which imposes self-consistency on the simulated data. Beyond these, comparisons to some of the many domain adaptation approaches to SBI from the literature would be in order.
- I find the literature review of relevant related work to be lacking. For example, the Autoencoding Variational Autoencoder (Cemgil et al.) has a similar notion of self-consistency applied to real data, but in the VAE setting. This line of work would be a promising area for future ablation studies that I think would improve the paper.

**Questions:**

- What precisely is meant by “sufficient statistics” in line 158? Sufficient in the formal statistical sense (if so, can you elaborate?), or do you mean more informally low-dimensional learned representations of $x$ that are known to have low reconstruction error, or something like that?
- The experiments appear to utilize a fixed simulation budget (e.g., 1024, line 723). Although this is the setting explored in Schmitt et al., my understanding of “standard” NPE typically allows unlimited samples from the model, e.g., fresh samples are drawn for each minibatch used to compute each gradient. I think the authors should also vary the simulation budget in their case study, as not doing so unfairly penalizes NPE (besides have the wrong model, it also has only finitely many samples from this model, and so will overfit).

---

> ### Author Response · Authors · 2025-11-20
>
> Dear Reviewer,
>
> We sincerely thank you for taking the time to evaluate our submission and for your thoughtful comments and suggestions. Please find our response below.
>
> **Weakness 1:**
>
> Thank you for initiating this discussion! Regardless of whether or not it applies to our current paper, we believe that just because a method combines existing methods into a new domain, it is not to be seen as an inferior contribution. Otherwise, almost all research should probably be dismissed as incremental. Our results show strong accuracy gains in out-of-distribution scenarios solving a major problem of neural amortized inference that was previously unsolved. As such, we firmly believe that it has major relevance for the field, also opening up several new avenues for future research to further improve the robustness and trustworthiness of neural amortized inference. For further details on our innovations, please see our responses below.
>
> **Weakness 2:**
>
> The self-consistency loss was proposed by Schmitt et al. 2024 to improve training efficiency for models with slow simulators in a **supervised setting**, that is, on simulated data only, where the goal lies in increasing simulation efficiency. Our work ports the idea to an entirely new problem domain, namely, ensuring that amortized posteriors are reliable under simulation gaps in an **unsupervised setting**, where true parameter values are unknown. Obtaining correct posterior estimates when the observed data is far away from the training data (i.e., in out-of-distribution scenarios) has been a long-standing problem in SBI and ABI, and to the best of our knowledge, no other approach allows to use unlabeled observed data to such a tremendous benefit.
>
> Also, through our empirical results, we found that minimising SC loss on a data different from simulated data achieves robustness even when the real observed data lies far away from both the training distributions.
>
> Additionally, we generalize the formulation and prove strict-properness of the SC loss which translates to SC loss always targeting the analytic posterior (Target 1, Elsemüller et al. 2025) independent of the input data. This is a novel approach to robustness in ABI as no previous work utilises it.
>
> **Weakness 3:**
>
> We respectfully note that some of our existing results may not have been fully apparent to the reviewer. Our experimental evaluations already incorporate most of reviewer’s suggestions as mentioned below:
>
> - We have not just compared our method with NPE but also benchmarked it against the state-of-the-art Stan (which uses NUTS, an advanced version of Hamiltonian Monte Carlo), providing highly accurate, but slow estimates
> - We have compared our method with three other recent robustness methods for ABI in Appendix D.1, including methods from domain adaptation.
> - We do compare with the method of Schmitt et al. in multivariate normal model case study (Appendix D, Figure 5, last row). The results show that utilising the method of Schmitt et al. under model misspecification does not improve robustness.
> - VAEs require large unlabelled data whereas SC shows strong robustness gains utilizing data in minimal amounts. The SC loss is strictly proper and its minimum corresponds to the true posterior. The VAE ELBO, in contrast, only guarantees a loose bound on data likelihood and does not enforce self-consistent Bayesian inference (see also our next response).
>
> **Weakness 4:**
>
> Thank you for this pointer. We will update the related work section to cite the paper by Camgil et al. We believe that the related work now covers all proposed robustness methods for ABI. Along with related work, Appendix D1 further explains additional unsupervised robustness methods, their inference targets, and how they compare with our method.
>
> While ‘‘consistency’’ has been explored in the context of autoencoding variational inference (Cemgil et al.), their formulation enforces encoder–decoder consistency on observed data through reconstruction-based ELBO optimization. In contrast, our self-consistency loss operates in the amortized Bayesian inference setting, where the goal is to ensure probabilistic consistency between the neural estimated posterior and the generative model. Unlike VAEs, our framework does not involve data reconstruction or a learned generative decoder, and thus addresses self-consistency at the level of Bayesian inference rather than variational autoencoding. Furthermore, our loss function does not minimize the ELBO but the expected forward KL instead.
>
> There are works which rely on minimising the distance between real and simulated data (Elsemüller et al. 2025), which would be closer to the consistency explored in Cemgil et al. However, these methods have a different inference objective and they perform poorly under prior misspecification where our method excels, as we demonstrate in Appendix D.1.

---

> > ### Author Response · Authors · 2025-11-20
> >
> > **Question 1:**
> >
> > By “sufficient statistics” we refer to low-dimensional representations of the data that contain all the information necessary for the posterior network to accurately infer the parameters. In other words, these learned summaries are “sufficient” for the purposes of posterior estimation, while discarding irrelevant variability. The sufficient statistics in the context of ABI are defined in Radev et al. 2020.
> >
> > **Question 2:**
> >
> > We did additional experiments for the multi-variate normal (Case study 1) and air traffic (Case study 2) where we trained NPE in online setting which allows fresh sample draws for each mini-batch. For NPE, the results show that while the posterior no longer collapses to point estimate when doing inference on data far from training distribution, the posteriors still remain biased. Similarly for air traffic case study, we train NPE in online setting for 40 epochs with a batch size of 32 and 32 batches per epoch, increasing the training set size from 1024 samples to 40960. However, the posterior estimates do not show any improvements and still remain biased. We will add these results in the camera-ready version.
> >
> > Here we report the Wasserstein distance from the online training results in air-traffic experiment:
> >
> > | Parameter | Wasserstein Distance (NPE)       | Wasserstein Distance (NPE+SC)        |
> > |-----------|------------------|----------------|
> > | $\alpha$        | 0.069 ± 0.020    | 0.026 ± 0.004  |
> > | $\beta$         | 0.105 ± 0.017    | 0.030 ± 0.005  |
> > | $\gamma$         | 0.156 ± 0.019    | 0.067 ± 0.013  |
> > | $\delta$        | 0.144 ± 0.024    | 0.064 ± 0.013  |
> > | $\log(\sigma)$   | 0.371 ± 0.038    | 0.039 ± 0.010  |
> >
> > **A comment on strengths:** We would like to clarify that self-consistency is not a ''regularizer'' but a strictly-proper loss function as we show in the Propositions and also mention this in lines 183-185.
> >
> > We hope that our responses adequately address the concerns and clarify any misunderstandings. We are happy to engage in further discussion if needed, and we respectfully hope that you will consider updating your evaluation in light of our clarifications.
> >
> > Best wishes,
> >
> > The authors
> >
> > **References:**
> > 1. Elsemüller, Lasse, et al. "Does Unsupervised Domain Adaptation Improve the Robustness of Amortized Bayesian Inference? A Systematic Evaluation." arXiv preprint arXiv:2502.04949 (2025).
> > 2. Radev, Stefan T., et al. "BayesFlow: Learning complex stochastic models with invertible neural networks." IEEE transactions on neural networks and learning systems 33.4 (2020): 1452-1466.

---

> > > ### Comment · Reviewer_Lkfz · 2025-11-23
> > > **Thank you for the detailed reply**
> > >
> > > Thanks for your reply. After reading the authors' and other reviewers responses, my original points remain unresolved. To summarize, these are 1) lack of novelty, 2) lack of illustration of the utility of the method, relative to existing work.
> > >
> > > Agreed with the authors that it's totally reasonable for new, meaningful work to be mostly combinatorial. However, in this case, as pointed out in my original review, the proposed objective seems to be **identical** to that of Schmitt et al., (2024), eq. 10, in the case where the simulator is well specified. To my reading, the self-consistency term in that objective does not require or utilize supervised data; it just depends on a given set of data $Y$ (see eq. 9 of Schmitt et al., 2024). In that work, $Y$ is simulated, here one has the availability of a real dataset. For me, this is already lacking on novelty, as the actual method proposed is to plug-and-play a different dataset into the same objective function.
> > >
> > > So to me, the authors' main contribution is a choice of problem setting where i) real, non-simulated data is available. Further, ii) they assume that the simulator is misspecified. However, the authors seem in their comparisons to want to restrict even to a further setting iii) where **the likelihood/model cannot be modified**, even though it is severely misspecified: this just seems unrealistic to me. In practice, I think one wouldn't use simulations at all in, for example, the case where real data typically has location 10 on the real line, but the simulator can only sample N(0,1) r.v.'s, as in S4.1. They would just fit a new likelihood model to their data, even if they only have a few points. And I don't think this is exactly an uncommon thing to do: VAEs, the original form of amortized Bayesian inference, typically fit the likelihood. I don't really buy in to the authors' assertion that "VAEs require large unlabelled data whereas SC shows strong robustness gains utilizing data in minimal amounts.", because the authors did not actually compare to a simple VAE. This ignores the quite accomplished field of research into likelihood estimation from real data going beyond VAEs.
> > >
> > > Also, there are other works that utilize both simulated and real data: the main line of work here I'm thinking of is the class of Reweighted Wake Sleep (Bornschein and Bengio) methods: a class of methods that uses both simulated (sleep) and real (wake) observations for amortized inference, as this work does, and seems like a natural competing method.

---

> > > > ### Author Response · Authors · 2025-11-24
> > > >
> > > > Dear Reviewer,
> > > >
> > > > We are unsure how to properly respond to your message since it seems you have already made up your mind. Let us nevertheless try to convince you to be more open to our arguments and contribution.
> > > >
> > > > You are right that we require (i) real data and that (ii) we are working in a setting where the simulator is not perfectly correctly specified (or insufficiently trained for other reasons). Point (ii) is simply the setup where we know neural posterior estimation struggles a lot, something that the field has recognized for several years now, without providing convincing solutions yet. We strongly believe SC is the right way forward, but as we can see from your responses, not everyone agrees with us. Point (i), the existence of real datasets, potentially many even, is the basic motivation for ABI in the first place, where the aim is to provide rapid inference not only on a single, but on many real datasets, providing several orders of magnitude faster inference than, say, MCMC.
> > > >
> > > > About point (iii) I think you are misunderstanding our paper. Yes, we do show *also* very severe misspecification to check the boundaries of our method, but this is in a toy example. The other case studies only investigate much milder misspecification in which standard NPE still yields highly unreliable results. And SC fixes that. We don’t know your scientific background, but in our background, we almost always deal with slightly misspecified models. And the goal is nevertheless to fit exactly the model we specified. If it doesn’t fit the data well, we will see this during post-processing based on which to then improve our model further (standard Bayesian workflow). And yet, for each model, we care about the posterior of that specific model given that specific data. It is what MCMC does too and who would argue against the widespread usefulness of MCMC, right?
> > > >
> > > > With regards to VAEs, these have not been the focus in SBI because they pursue a very different goal: learn an amortized posterior over **unknown** latent variables. In SBI, we are not learning the latent variables, since these are **known** through the mathematical model and we only care about **accurate posterior estimation**. Thus, the setup we target is the most standard one in SBI, as  evidenced by hundreds of papers. Moreover, even though a classical encoder-decoder VAE is, in principle, applicable in the SBI setting, the field converged on normalizing flows (and more recently, diffusion models) years ago, since these can be interpreted as bijective VAEs.
> > > >
> > > > Best wishes,
> > > >
> > > > The authors

---

> > > > > ### Comment · Reviewer_Lkfz · 2025-11-25
> > > > > **Rephrasing**
> > > > >
> > > > > Thanks for the reply. I think some of my main points above could be articulated more clearly so that we are on the same page, so I want to clarify to give the authors a chance to address my main points.
> > > > >
> > > > > 1) I don't think any reviewers, myself included, are worried about comparisons to MCMC. We all agree that amortized inference is very useful.
> > > > >
> > > > > 2) **The authors have not yet addressed in their responses the fact that the proposed objective utilized is identical to that of Eq. 10 of Schmitt et al. (2024).** This is one of my largest concerns regarding novelty on this submission, which I don't think the authors have engaged with yet. The authors also did not compare to Schmitt et al., (2024) in their experiments. They compare to NPE, which uses only simulated data. Schmitt et al., (2024) also uses only simulated data and has the same objective function, so this is a natural baseline that is missing.
> > > > >
> > > > > 3) By VAE, both myself and other reviewers are simply referring to optimizing the amortized ELBO objective. Precisely, this objective is given by
> > > > > $L(\psi,\phi) =
> > > > > E_{x \sim Data} E_{\theta \sim q_\phi(\theta \mid x)}
> > > > > \left[ \log \frac{p_\psi(x,\theta)}{q_\phi(\theta \mid x)}\right]
> > > > > $
> > > > > in parameters $\psi, \phi$. This can be used in any amortized inference problem where some data and a likelihood model are available, over a known latent variable of interest $\theta$, that is explicitly modeled upfront. Alongside learning the inference network, one fits the model $p_\psi(\theta,x)$ as well. I would actually argue this is the most common setting, to fit $p$ and $q$ jointly.
> > > > >
> > > > > 4) The key issue is that the authors have access to a likelihood function and real data, but **do not evaluate against comparable methods**. None of NPE, NPE-DANN, NPE-MMD, and NNPE have access to a likelihood function, so their underperformance is natural.
> > > > >
> > > > > 5) **Fair evaluation would allow for fitting the model likelihood.** Especially if the model is known to be misspecified, as in this case. In related work on amortized Bayesian inference, the likelihood/model $p_\psi(\theta, x)$ is almost always fit to the data, if real data are available. As the authors point out above: "If it doesn’t fit the data well, we will see this during post-processing based on which to then improve our model further (standard Bayesian workflow)". Yet, the proposed method does not allow any way to fit the model. As the authors also point out, nowadays with normalizing flows and diffusion models, $p_\psi$ can match the data extremely well.
> > > > >
> > > > > 5) **Fair evaluation would compare to other approaches with access to a likelihood function.** In addition to comparing to the amortized ELBO, I think the authors have to compare to other objective functions in literature that use either a notion of self-consistency or a mix of simulated and real data. Omitting these is a significant gap. These include eq. 10 of [1], and eq. (3) and (5) of [2], among others. In particular, since using a mix of simulated and real data is posited as the main innovation, I think comparisons to Reweighted Wake-Sleep [2] and that line of work have to be included, as this is the main line of work that uses both simulated and real data for amortized inference.
> > > > >
> > > > > [1] The Autoencoding Variational Autoencoder. Cemgil et al.
> > > > >
> > > > > [2] Reweighted Wake Sleep. Bornschein and Bengio.

---

> ### Author Response · Authors · 2025-11-26
>
> Dear Reviewer,
>
> Thank you for rephrasing your questions. Please find our response below.
>
> **Question 2:**
>
> **We did in fact compare our method against Schmitt et al. 2024 in Figure 5, last row, Appendix D, and the results clearly show that their method does not overcome model misspecification**, in the sense that it provides results not more accurate than NPE alone. To further strengthen our point, we did additional experiments using the method of Schmitt et al. on our second case study (air traffic) and we show the results below. As you can see, Schmitt et al. does not improve accuracy of the approximate posterior in the misspecification regime (it showed similar accuracy to NPE alone).
>
> WD: Wasserstein distance w.r.t Stan
>
> | Parameter | WD (NPE) | WD (NPE+SC) | WD Schmitt et al. |
> |-----------------------------------------------------------|-----------------------------|-------------------------------|-----------------|
> | $\alpha$                                                | $0.080 \pm 0.028$         | $0.010 \pm 0.002$           | $0.105 \pm 0.030$ |
> | $\beta$                                                 | $0.088 \pm 0.020$         | $0.018 \pm 0.002$           | $0.113 \pm 0.022$ |
> | $\gamma$                                                | $0.113 \pm 0.015$         | $0.018 \pm 0.003$           | $0.204 \pm 0.026$ |
> | $\delta$                                                | $0.110 \pm 0.020$         | $0.017 \pm 0.002$           | $0.206 \pm 0.028$ |
> | $\log \sigma$                                           | $0.227 \pm 0.045$         | $0.016 \pm 0.002$           | $0.547 \pm 0.073$ |
>
> About your question of conceptual differences, Schmitt et al. 2024 introduced the self-consistency loss and showed it improves data efficiency in low-data, simulation-based settings. Whereas we generalize that idea to a semi-supervised ABI framework that uses unlabeled (real-world) data, prove strict-properness of the semi-supervised losses, and demonstrates strong out-of-simulation robustness. When utilising simulated data in Eq. (3) of our paper, it becomes identical to Eq. (10) of Schmitt et al. 2024, but the novelty of our approach lies in recognising that self-consistency loss is unsupervised and adapting it for improving robustness under misspecification utilising unlabeled (real-world) data. Our empirical results also show that SC achieves robustness gains even when utilising as few as four unlabeled observations, thus making it the first few-shot semi-supervised approach to robust ABI.
>
> **Question 3:**
>
> We conducted additional experiments using a VAE baseline with a known decoder (likelihood) and a neural encoder (posterior network). The encoder is a feed-forward network with three hidden layers of 128 units each, followed by two output heads that parameterize the mean and standard deviation of a standard Normal base distribution. The model was trained on 40,960 simulated pairs until convergence. We trained NPE and NPE + SC again in online setting that leads to training data of the same size as VAE. Results for the air-traffic experiment (Case Study 2) are reported in the table below:
>
> WD: Wasserstein distance w.r.t Stan
>
> | Parameter      | WD (VAE)            | WD  (NPE)           |  WD (NPE+SC)          |
> |----------------|------------------------------|------------------------------|-------------------------------|
> | $\alpha$     | $0.065 \pm 0.016$          | $0.080 \pm 0.028$          | $0.010 \pm 0.002$           |
> | $\beta$      | $0.102 \pm 0.019$          | $0.088 \pm 0.020$          | $0.018 \pm 0.002$           |
> | $\gamma$     | $0.125 \pm 0.019$          | $0.113 \pm 0.015$          | $0.018 \pm 0.003$           |
> | $\delta$     | $0.128 \pm 0.027$          | $0.110 \pm 0.020$          | $0.017 \pm 0.002$           |
> | $\log \sigma$| $0.144 \pm 0.024$          | $0.227 \pm 0.045$          | $0.016 \pm 0.002$           |
>
> These results show that the performance of VAE with known likelihood is comparable to NPE (which does not use explicit likelihood density) but it is much worse than our semi-supervised (NPE + SC) approach. We will include these results in the camera-ready version.
>
> **Question 4:**
>
> All of the baseline robustness methods we compare against (NNPE, NPE-MMD, NPE-DANN) are designed specifically for utilising unlabeled real-world data (as our SC loss), which would be a fair comparison from that perspective. To the best of our knowledge, no existing ABI robustness method other than ours leverages the explicit likelihood density, but it can hardly be held against our method that other such methods have not yet been developed. We also compare to the method of Schmitt et al. and VAE which use the likelihood density too, but the results show that these methods do not improve robustness.

---

> > ### Author Response · Authors · 2025-11-26
> >
> > **Question 5:**
> >
> > Your last sentence suggests that you have some misconceptions about how neural ABI works. In ABI, the inference network (e.g., a normalizing flow or diffusion model) *does not* fit the data or the likelihood model itself, but rather approximates the posterior distribution of the statistical model that in turn is used to fit the data. So the flexibility of the inference network does not affect how well the statistical model fits the data. These are two different aspects (the statistical model vs. the posterior approximator), as for example explained in Bürkner et al. 2022[1]. Our work targets the robustness of the posterior approximator without modifying the likelihood, which is a highly important scenario in Bayesian modeling.
> >
> > **Question 6:**
> >
> > While interesting parallels could be drawn between our semi-supervised self-consistency formulation and Cemgil et al., integrating such comparisons would be beyond the scope of current paper and could be a project on its own. AVAE’s objective tries to overcome encoder/decoder inconsistency through latent parameters, but in ABI, the posterior network is already trained on simulated pairs ($\theta, x$), that is, the core AVAE fix (enforcing generated samples map back) is largely already present in the NPE loss. Our focus in this work is on full-fledged ABI evaluation, and we therefore compare against existing robustness-oriented ABI baselines (NNPE, NPE-DANN, NPE-MMD, and Schmitt et al. 's supervised SC).
> >
> > The Reweighted Wake–Sleep algorithm fundamentally differs from our setting because it updates the generative model (i.e., the likelihood) during training. Whereas our work targets the robustness of the posterior approximator **without modifying the likelihood**. Therefore, it won’t be a fair comparison.
> >
> > Best wishes,
> >
> > The Authors
> >
> > [1] Bürkner, Paul-Christian, Maximilian Scholz, and Stefan T. Radev. "Some models are useful, but how do we know which ones? Towards a unified Bayesian model taxonomy." arXiv preprint arXiv:2209.02439 (2022).

---

### Official Review · Reviewer_bLfT · 2025-11-01

**Soundness:** 3
**Presentation:** 3
**Contribution:** 3
**Rating:** 8
**Confidence:** 3

**Summary:**

This paper introduces self-consistency loss proposed to improve training efficiency into amortized Bayesian inference (ABI) to enhance its robustness. In addition to the loss based on labeled simulation data, it includes the self-consistency loss defined with unlabeled real data. The later adds extra information to improve the robustness of ABI.

**Strengths:**

The independence of true parameter values is the most striking feature of the proposed method. The method is justified by theories. Various numerical evidences including through simulation and real-data applications are strong and convincing.

**Weaknesses:**

The conditions of propositions 2 and 3 are not clear to me (See question below). Figure 4 (b) is a little bit misleading. Overall, these are minor defects.

**Questions:**

1. Does the condition of proposition 2 always hold? Does proposition 3 depend on on proposition 2? If so, the statement of proposition 3 reads like it holds regardless. Some clarification will be appreciated.

2. Figure 4 -(b) is misleading. I understand that it represents the difference between MAB (NPE) and MAB (NPE+SC). However it looks that MAB (NPE) is 0 while MAB (NPE+SC) is bigger and concentrated around 5. Why not plotting the histograms of each using different colors?

---

> ### Author Response · Authors · 2025-11-20
>
> Dear Reviewer,
>
> We sincerely thank you for taking the time to evaluate our submission and for your thoughtful comments and suggestions. Please find our response below.
>
> **Response to Weakness and Question 1:**
>
> Proposition 3 guarantees strict properness of the semi-supervised objective provided the self-consistency score $C$  satisfies the condition of Proposition 1 (i.e., is globally minimized iff its argument is constant across the posterior support). Prop. 3 holds for every instantiation of $C$  that satisfies Prop. 1 and it does not depend on Prop. 2.
>
> Proposition 2 gives a concrete sufficient condition showing that the variance-based $C$ (Eq. (4)) meets Proposition 1 only when the proposal $ p_C (θ)$ has support that encompasses the support of  $p(θ \mid x)$. Thus, when using the variance self-consistency loss, the practitioner must ensure a choice of  $p_C$   that covers the posterior support.
>
> In the camera-ready version, we will further clarify the relations of Propositions 1 to 3 to make their assumptions more clear.
>
> **Response to Question 2:**
>
> That’s a good idea! In the camera-ready version will we change the plot accordingly.
>
> We hope that our responses adequately address the concerns and clarify any misunderstandings. We are happy to engage in further discussion if needed, and we respectfully hope that you will consider updating your evaluation in light of our clarifications.
>
> Best wishes,
>
> The authors

---

### Meta-Review · Area_Chair_gy78 · 2026-01-04

**Summary:**

Amortized Bayesian inference with neural networks enables fast solutions to inverse problems but suffers from severe robustness issues when test data is outside of the training distribution. This paper introduces a propose a self-consistency loss based semi-supervised approach allowing the use of unlabeled and real data. Experiments on time-series and image-based inverse problems show substantially improved robustness and accurate inference in out-of-simulation regimes.

**Reviewer Concerns:**

Reviewers main concerns were:
1. insufficient novelty compared to Schmitt at al that suggested the self-consistency loss. The authors made clear, that they reused the self-consistency loss but brought it to a totally different setting.
2. insufficent evaluation: method should be compared against that of Schmitt at all and a VAE. Almost all of the experiemtns the reviewer asked for were already presented in the paper or the appendix, and additional suppoertive experiments were presented in the rebuttal.
3. lacking references in the literature review, which were added by the authors.
4. missing clearaty of theoretical guarantees. The authors could clearify this points in the rebuttal.
5. practical relevance of the analysed problem unclear.  The authors convincingly argued that ABI targets fast posterior approximation under a fixed model, even when the likelihood is known, and that model misspecification is common and unavoidable in practice.

 The rebuttal solved the most serious concerns.

**Reviewer Scores:**

Reviewer bLfT already voted for acceptance (score 8) and had only two minor comments/questions that were clearified during the rebuttal.
Reviewer  Lkfz is the only one that voted for rejection  (score 2) and his main reason is the missing novelty from his view. All other concerns were convincingly discused in the rebuttal.
Reviewer L5ck voted for accepane (6) already before the rebuttal and his concerns were nicely discussed, so I do not assume that he would have lowerd his score.
Reviewer ipsv indicated that they would incease their score (from 4).

Overall I think the paper would have a score over 5 after rebuttal, and the main reason for the negative review should not be taken too much into account.

---

### Decision · Program_Chairs · 2026-01-26

Accept (Poster)